

# Integrating Agricultural Practices into the TRIPLEX-GHG Model v2.0 for Simulating Global Cropland Nitrous Oxide Emissions: Model Development and Evaluation

5   Hanxiong Song[1,2], Changhui Peng[1,2]*, Kerou Zhang[3], and Qiuan Zhu[4]

1.Department of Biology Sciences, University of Quebec at Montreal, C.P. 8888, Succ. Center-Ville, Montreal H3C 3P8, Canada

2. Center for Ecological Forecasting and Global Change, College of Forestry, Northwest A&F University, Yangling, China

3. Institute of Wetland Research, Chinese Academy of Forestry, Beijing, China

10   4. College of Hydrology and Water Resources, Hohai University, Nanjing, China

*Correspondence to*: Changhui Peng (peng.changhui@uqam.ca)



**Abstract** Nitrous oxide ($N_2O$) emissions from croplands are one of the most important greenhouse gas sources, and it is difficult to simulate on a large scale. In order to simulate $N_2O$ emissions from global croplands, a new version of the process-based TRIPLEX-GHG model was developed by coupling the major agricultural activities. The coefficient of the $NO_3^-$ consumption rate for denitrification ($COE_{dNO3}$) was found to be the most sensitive parameter based on sensitivity analysis, and it was calibrated using field data from 39 observation sites across major croplands globally. The model performed well when simulating the magnitude of the daily $N_2O$ emissions and was able to capture the temporal patterns of the $N_2O$ emissions. The $COE_{dNO3}$ ranged from 0.01 to 0.05, and the continental mean of the parameter was used for the model validation. The validation results indicate that the means of the measured daily $N_2O$ fluxes during the experiment periods are highly correlated with the modeled results ($R^2 = 0.87$). Consequently, our model simulation results demonstrate that the new version of the TRIPLEX-GHG model can reliably simulate $N_2O$ emissions from various croplands at the global scale.

**Key Words: $N_2O$, croplands, process-based model, fertilization**



## 1. Introduction

25  Nitrous oxide ($N_2O$) is a long-lived trace gas that has a global warming potential on a 100-year time horizon that is 265–

298 times larger than that of carbon dioxide ($CO_2$), and it simultaneously results in ozone depletion in the stratosphere (Ciais

et al., 2014). The atmospheric concentration of $N_2O$ has significantly increased (i.e., by 20%) since the industrial revolution

(Tian et al., 2016; Tian et al., 2020). Generally, $N_2O$ is produced as an intermediate product of soil microbial nitrification and

denitrification processes and is regulated by multiple biotic (i.e., vegetation type, microbial biomass) and abiotic factors (i.e.,

climate, soil temperature, humidity, nutrient content, and texture) (Bouwman et al., 2002; Stehfest and Bouwman, 2006; Li et

al., 2000; Butterbach-Bahl et al., 2013; Tian et al., 2018).

Cropland is a hotspot of terrestrial $N_2O$ sources (Tian et al., 2019; Tian et al., 2020). The current larger emission rate of

cropland soil comparing with natural soil (Davidson and Kanter, 2014) results from extensive agricultural practices, including

N-fertilizer input (synthetic and manure) (Davidson, 2009; Zhou et al., 2017), irrigation (Li et al., 2000; Li et al., 2010), and

tillage (Powlson et al., 2014; Mei et al., 2018), because these agricultural practices directly and indirectly interfere with soil N

flow and microbial activities (Cavigelli et al., 2012). Therefore, substantial observation studies have been conducted in

croplands to understand the effects of different agricultural practices on $N_2O$ emissions in order to enable sustainable

agricultural production (Carlson et al., 2017; Burney et al., 2010; Snyder et al., 2009). However, because of the characteristics

of the varying magnitudes across the study sites and periods (Tian et al., 2016; Tian et al., 2019), the emission pattern of $N_2O$

requires models to be quantitatively investigated on large scales (Li et al., 2000; Wrage et al., 2001; Tian et al., 2018).

Modeling is an important approach for quantifying the $N_2O$ emissions from various ecosystems, especially croplands, under

changing environments and management. Linear and non-linear models based on emission factors (EF) have been widely used

to estimate direct $N_2O$ emissions on different scales (Shcherbak et al., 2014; Davidson, 2009; Gerber et al., 2016; Hoben et al.,

2011). However, the EF method has been questioned due to the large uncertainty generated by its inability to depict spatial

(i.e., site, regional and global) and temporal (i.e., monthly, daily) variations (Ehrhardt et al., 2018; Tian et al., 2019; Berdanier

and Conant, 2012). Models based on machine learning algorithms such as the random forest algorithm (Philibert et al., 2013),

artificial neural network (Oehler et al., 2010), and Bayesian inversion (Berdanier and Conant, 2012) have recently been applied

to cropland $N_2O$ emission estimations, but these methods strongly depend on the quality of the training data, instead of the

underlying mechanism of the $N_2O$-related processes.

Process-based biogeochemical models, which serve as an alternative, have been demonstrated to be an effective tool for

assessing and predicting the $N_2O$ flux by describing the $N_2O$ emission processes and dynamics and by integrating the natural

and anthropogenic drivers at different scales (Tian et al., 2019; Tian et al., 2018). The DAYCENT (Daily Century) model has

provided adequate simulations of $N_2O$ fluxes for a variety of agroecosystems with different scales (Del Grosso et al., 2005;

Cheng et al., 2014; Del Grosso et al., 2009; Alvaro-Fuentes et al., 2017). Nevertheless, because it predominately utilizes simple




functions based on soil water, inorganic nitrogen (N) concentrations, respiration, and texture (Del Grosso et al., 2000; Parton et al., 1996), the limited model descriptions for oxygen diffusion and consumption processes lead to relatively large uncertainties, especially for disturbed soils (Butterbach-Bahl et al., 2013; Alvaro-Fuentes et al., 2017; Song et al., 2019). The Carnegie-Ames-Stanford (CASA) biosphere model estimates the soil $N_2O$ flux based on the concept of the "hole in a pipe

model" (Potter et al., 1996), but the lack of a detailed description of the microbial activities limits the model's performance (Li et al., 2020). Tian et al. (2010) developed a process-based biogeochemical model, i.e., the Dynamic Land Ecosystem Model (DLEM), which has been successfully used to estimate $N_2O$ emissions at continental and global scales (Tian et al., 2010; Xu et al., 2017). However, due to the absence of the effect of soil pH, the nitrification and denitrification processes were simulated based on empirical models (Chatskikh et al., 2005; Heinen, 2006), which might be responsible for the bias of the modeled

results. The DeNitrification-DeComposition (DNDC) model, developed by Li et al. (1992), which is a well-known process-based model, has been widely used to estimate $N_2O$ emissions and crop production in agroecosystems on site to regional scales (Li et al., 2000; Giltrap et al., 2010; Lugato et al., 2010). However, the proper application of the DNDC requires relatively complex agricultural practices as input information, which limits its large-scale modeling ability (Perlman et al., 2014). Furthermore, dynamic global vegetation models (DGVM) have also been coupled with $N_2O$-related processes. Because of the

advantage that they reflect the vegetation response to climate change, DGVMs are capable of simulating $N_2O$ emissions on a global scale (Saikawa et al., 2013; Xu and Prentice, 2008; Xu et al., 2012; Zhang et al., 2017b). For instance, Xu-Ri et al. (2012) successfully developed the DyN-LPJ model to estimate the total $N_2O$ emissions from the global terrestrial ecosystem. However, by only integrating simple semi-empirical equations without the complex, subsidiary processes of the $N_2O$ dynamics, the DyN-LPJ model has not been used to simulate $N_2O$ emissions from fertilized agricultural soils or soils with any other

anthropogenic effects (Xu et al., 2012; Xu and Prentice, 2008).

The global $N_2O$ Model Intercomparison Project (NMIP) has compared the modeled $N_2O$ emissions from global terrestrial ecosystems simulated using 10 process-based biogeochemical models, and it has been reported that large uncertainties still exist in the current estimations of the global $N_2O$ budget (Tian et al., 2018). These results have been confirmed by ensemble model studies (Ehrhardt et al., 2018; Tian et al., 2019), and the modeled uncertainties are probably generated by the large

temporal and spatial variations in the $N_2O$ flux and the differences in the model structures, parameterization schemes, and input datasets. Therefore, further improvement of process-based $N_2O$ emission models is critical for reducing the global modeling uncertainties and for closing the global $N_2O$ budget in order to cope with the global change.

As a recently developed process-based model, the TRIPLEX-GHG (Zhu et al., 2014) can simulate multiple ecological processes and has been successfully applied to simulate $N_2O$ fluxes from natural ecosystems (grasslands, forests) (Zhang et

al., 2017b). However, the impact of human disturbances (e.g., agricultural practices, land use changes, and management) have not been considered so far (Tian et al., 2018). In this study, we enhanced the TRIPLEX-GHG model's capability by addressing





the impacts of major agricultural practices on the N$_2$O production and emission processes in order to simulate N$_2$O emissions from global croplands. The objectives of this study were: (1) to develop agricultural practice modules in the framework of an extant process-based model (i.e., the TRIPLEX-GHG); (2) to conduct a sensitivity analysis to identify the most sensitive parameter; and (3) to validate modeled the results using field observations of various cropland sites at the global scale.

## 2. Model description

The TRIPLEX-GHG model (Peng et al., 2013; Zhang et al., 2017b; Zhu et al., 2014) is a process-based terrestrial ecosystem model, which is based on the Integrated Biosphere Simulator (IBIS) (Foley et al., 1996; Kucharik et al., 2000) and TRIPLEX (Peng et al., 2002). The basic structure of the original TRIPLEX-GHG model and the integration of agricultural management processes are shown in Fig. 1, and are described in detail below.

The TRIPLEX-GHG model consists of four key submodules: a land surface submodule for simulating the energy budget and hydrological cycle between the soil surface, vegetation canopy, and the atmosphere; a dynamic vegetation submodule that is used to determine the geographic distribution of specific plant functional types (PFTs) under climate change; a plant phenology submodule that describes the dominate phenological behavior of each PFT based on a set of phenological parameterizations (Botta et al., 2000); and a soil biogeochemical submodule that simulates the dynamics of the C and N flows and the major microbial processes, including nutrient mineralization, immobilization, and their interactions with the environment. Specifically, the biogeochemical processes mostly focus on the C cycle within three plant biomass pools (leaf, root, and wood, each of which can be further divided into the metabolic, structural, and lignin pool) and three soil organic matter pools (litter, humus, and microbial), which are comprised of non-protected, protected, and passive organic matter. However, the N cycle's scheme is coupled with the C cycle and relies on the corresponding C:N ratios of the different organic matter pools and two additional inorganic N pools (nitrite-N [NO$_3^-$] and ammonium-N[NH$_4^+$]).

By incorporating the decomposition, methane (CH$_4$) production and oxidation, nitrification, and denitrification processes with the original model, the TRIPLEX-GHG model has been validated, modified, and used to simulate major green-house gas emissions from natural terrestrial ecosystems (grasslands, forests, and wetlands) (Zhang et al., 2017b; Zhang et al., 2019; Zhu et al., 2014; Zhu et al., 2015). However, the current TRIPLEX-GHG model does not include major agricultural practices, and thus, it is unable to accurately simulate the N$_2$O flux from agricultural soils. To overcome this problem and reduce the uncertainties in global N$_2$O simulations, the main framework for improving the TRIPLEX-GHG model was to add a new component that takes into account how agricultural practices affect the biogeochemical cycle, especially the nitrification and denitrification processes, thus modifying the pattern of the N$_2$O flux of croplands at the global scale.

### 2.1 The N$_2$O module



As a trace N gas, the $N_2O$ emitted by nitrification and denitrification were simulated according to the anaerobic balloon concept. The anaerobic volumetric fraction (ANVF) was the key parameter, which represents the soil oxygen status and

regulates the allocation rates of the substrates (e.g., dissolved soil organic carbon (DOC), $NH_4^+$, and $NO_3^-$) for nitrification and denitrification. It was calculated using the oxygen partial pressure and the air-filled porosity listed in Supporting Information Table S1 (Equations (1–3)).

In the TRIPLEX-GHG model, nitrification is an aerobic process that occurs outside of the anaerobic balloon, converting ammonium ($NH_4^+$) into nitrate ($NO_3^-$) driven by nitrifying bacteria with $N_2O$ as a by-product (Li et al., 2000; Zhang et al.,

2017b; Morkved et al., 2007). The growth and death rate of nitrifiers (Equations (4–6) in Table S1) are highly dependent on the DOC; therefore, the nitrification rate (Equations (7–10) in Table S1) was calculated using the Michaelis–Menten function based on the concentration of $NH_4^+$ and the microbial activity of the nitrifying bacteria. The effects of the soil properties were also simulated (Equations (11–12) in Table S1).

Denitrification is the process through which the nitrate is reduced stepwise into different nitrogen gases as a chain reaction

process inside of the anaerobic balloon. Denitrification can be divided into 4 independent steps, which are linked by the competition for DOC by the specific denitrifiers during each step (Betlach and Tiedje, 1981). Similarly, the growth and mortality rates of the different denitrifiers utilized a double substrate based (DOC and NOx) Michaelis–Menten equation (Equations (13–14) in Table S1). The consumption of $NO_X$ for the growth of the different denitrifiers was calculated at an hourly time step according to previous studies as is shown in the following Eq. (1) (Leffelaar and Wessel, 1998; Li et al., 2000):

$$F_{ANNOX} = COE_{dNOX} \cdot B_{denit} \cdot \left( \frac{R_{NOX}}{EFF_{NOX}} + \frac{MAI_{NOX} \cdot [NO_X]}{[N]} \right) \cdot f_{NOX}(pH) \cdot f(t). \qquad (1)$$

Here, $F_{ANNOX}$ is the consumption rate of $NO_X$ (kg N m$^{-3}$ h$^{-1}$); $COE_{dNOX}$ is the coefficient of $NO_X$ consumption; $B_{denit}$ is the biomass of the denitrifiers (kg C m$^{-3}$); $R_{NOX}$ is the $NO_X$ reduction rate (h$^{-1}$); $EFF_{NOX}$ is the efficiency of the $NO_X$ denitrifiers (kg C kg N$^{-1}$); $MAI_{NOX}$ is the maintenance coefficient of $NO_X$ (h$^{-1}$); $[NO_X]$ and $[N]$ are the $NO_X$ and total N concentrations in the anaerobic balloon, respectively; and $f_{NOX}(pH)$ and $f(t)$ are the effects of the soil pH and soil

temperature on the $NO_X$ denitrification rate in each step, respectively (Equations (15–17) and Equation (18) in Table S1).

In our model, all of the gaseous products of the nitrification and denitrification ($N_2O$ primarily) are emitted from the bottom layer of the soil into the atmosphere primarily driven by diffusion based on the $N_2O$ concentration and soil depth according to Fick's law of diffusion (Equations (19–20) in Table S1).

### 2.2 Effects of agricultural management practices

$N_2O$ production and emission are key parts of the soil N cycle. These processes are controlled by the environmental conditions, which directly respond to the varying soil management practices (Butterbach-Bahl et al., 2013). Understanding the direct and indirect effects of the different agricultural practices on the soil N flow (N input and output) is critical to accurately predicting the $N_2O$ fluxes in cropland ecosystems (Liu et al., 2010).



### 2.2.1 N output

Except for gaseous N losses (N$_2$O, NO, NH$_3$), crop uptake/harvest processes, leaching, and surface runoff are the major N outputs (Liu et al., 2010). We altered the plant N uptake process and integrated harvest practices into the original model.

Plant N uptake: As a plant grows, mineral N is taken up as NO$_3^-$-N and NH$_4^+$-N, which is considered to be the dominant pathway of soil N loss (Sebilo et al., 2013). It has been reported that NO$_3^-$-N is much more easily absorbed by roots (Malhi et al., 1988; Kronzucker et al., 1997), so NO$_3^-$-N was set as having a higher priority of being taken up by plant roots in each soil

layer using the following Eq (2) and Eq (3):

$$demand\_NO_{3\ i}^- = COE_{NO3} \cdot layer\_demand_i \cdot \frac{NO_{3\ i}^-}{(NO_{3\ i}^- + NH_{4\ i}^+)}, \qquad (2)$$

$$demand\_NH_{4\ i}^+ = layer\_demand_i - demand\_NO_{3\ i}^-. \qquad (3)$$

Here, $demand\_NO_{3\ i}^-$ and $demand\_NH_{4\ i}^+$ are the plants' NO$_3^-$ and NH$_4^+$ requirements from soil layer $i$; and $layer\_demand_i$ is the total plant N uptake requirement from soil layer $i$. $COE_{NO3}$ is the coefficient of nitrate demand, which

was set to 4.0 according to the model test. The comparison between the tendency of the modeled soil nitrate content and the detected soil nitrate concentration proved the effectiveness of the model design (Fig. S1).

Harvest: Harvest practices significantly reduce the soil C and N inputs for cropland compared with natural soil. We systematically removed all of the litterfall from the cropland ecosystem at the end of the growing seasons to modify the harvest.

### 2.2.2 N Input

The input N flows of agricultural soil include N fertilizer, biological N fixation, atmospheric deposition, and returned straw (Liu et al., 2010). We integrated chemical N fertilization, manure application, and returned straw processes into the original model.

Returned residues: Returned residues are a significant C and N source for cropland soil and a recommended practice for improving N use efficiency. It has been reported that more than half of the N in crops (all of which is taken up from the soil) is

removed from the ecosystem, and only 20% of the crop's N is returned to the soil N pools as returned residues globally (Liu et al., 2010). Therefore, in our model, we returned 20% of the harvested plants to the cropland soil in order to modify the returning residue practices.

Chemical N fertilizer: Chemical N fertilizers were directly added to the NO$_3^-$ and NH$_4^+$ pools of the top layer of soil in the original model.

Manure N fertilizer: The manure-sourced N entered the different inorganic N and organic N pools separately. The organic portion of the manure was added to up to 3 soil organic matter (hereafter SOM) pools (the non-protected, protected, and passive organic carbon pools) separately for further decomposition (Zhang et al., 2017b).

$$ManureNH_4^+ = R_{NH4} \cdot Manure_N. \qquad (4)$$

$$ManureNO_3^- = R_{NO3} \cdot Manure_N. \qquad (5)$$





$$ManureC_{SOM} = proportion_{SOM} \cdot C{:}N_{SOM} \cdot Manure_N. \tag{6}$$

Here, $ManureNH_4^+$ and $ManureNO_3^-$ are manure-sourced $NH_4^+$-N and $NO_3^-$-N, respectively, which are calculated using the ratio of ammonia and nitrate (i.e., $R_{NH4}$ and $R_{NO3}$) to total manure N. $ManureC_{SOM}$ is the amount of manure that entered the different SOM pools; $proportion_{SOM}$ is the proportion of manure N added to the different SOM pools; and $C{:}N_{SOM}$ is the C:N ratio of a particular SOM pool.

**2.2.3 Irrigation and tillage**

Some agricultural practices do not directly alter the N flow in cropland soil, instead they affect $N_2O$ by regulating the soil's physical properties.

Irrigation: The irrigation process used in this study adopted the idea of precipitation events from the DNDC (Li et al., 2000) and Agricultural Production Systems sIMulator (APSIM) (Thorburn et al., 2010). Similar to rainfall, irrigation provides extra

water to the surface of the cropland soil, promoting the water-filled pore space (hereafter WFPS), thus stimulating the growth of the anaerobic balloon. Nevertheless, it simultaneously induces leaching and runoff processes. In the current model, only the flood irrigation method was included.

Tillage: Tillage redistributes the soil profile and increases the availability of oxygen in each soil layer at the same time. We averaged each of the C and N pools of the top 3 soil layers (as a global conventional tillage depth) after every tillage event.

Because of more exposure to oxygen, the anaerobic conditions and diffusion pattern also vary with the different soil moisture conditions, properties, and vegetation types (Rochette, 2008; van Kessel et al., 2013).

**3. Data and methods**

**3.1 Model sensitivity analysis**

We conducted initial sensitivity analysis experiments to obtain the most sensitive parameters before testing the model. According to previous $N_2O$ modeling studies (Zhang et al., 2017b; Zhang et al., 2019), the coefficient of nitrification (hereafter $COE_{NR}$) is the key parameter driving the amount of emitted $N_2O$ in natural ecosystems probably because of the limited $NO_3^-$ input. In this study, considering the increased $NO_3^-$ input from fertilizers in cropland soil, it is conceivable for denitrification to become the dominant $N_2O$ source. Therefore, 13 major parameters, including $COE_{NR}$ and parameters associated with

denitrification (Table1), were compared in a site-specific manner. The sensitivity index (SI) in this study followed the method of Lenhart et al. (2002) using the following Eq(7):

$$SI = \frac{1}{n} \cdot \sum_{j=1}^{n} \left( \frac{(y_{2j} - y_{1j})/y_{0j}}{2 \cdot \Delta x / x_0} \right), \tag{7}$$

where n is the total number of months from 1961 to 2015 (because in our model, chemical fertilizer application started in 1961); j accounts for the number of months from 1961 to 2015 (because in our model, chemical fertilizers were used after

1961); $y_{0j}$ represents the jth monthly $N_2O$ emissions with an initial parameter $x_0$; and $y_{2j}$ and $y_{1j}$ are the $N_2O$ emission





values produced for $+\Delta x$ and $-\Delta x$, respectively. $\Delta x$ was set as 20% of $x_0$.

### 3.2 Model input data, calibration, and validation

### 3.2.1 Studied sites

We compiled measured $N_2O$ emission data from croplands in published studies and the locations of the selected sites were distributed across most of the dominant terrestrial area. Most of the major crop types (maize, wheat, corn, sugarcane, vegetables, and cotton) were represented. The detailed site information is listed in Table 2, including the geographic location (latitude, longitude, and specific site), experimental period, dominate crop type, average N dose, soil properties (soil organic carbon, hereafter SOC, soil pH, soil texture), average daily $N_2O$ emissions during the experimental period, and other agricultural

practice information. Table S2 provides similar information on the dataset used for the validation.

### 3.2.2 Input data

All of the input information for the model simulation of the selected sites described above was directly obtained from the following datasets or was obtained from papers (see details below). These data were transformed into a spatial resolution of $0.5° \times 0.5°$ latitude/longitude using the ArcMap software (version 10.2) before the simulation.

**Daily climate data:** We obtained daily climate data from the CRUNCEP dataset (https://www.earthsystemgrid.org/dataset/ucar.cgd.ccsm4.CRUNCEP.v4.TPHWL6Hrly.html), including the minimum, average, and maximum temperature, precipitation, specific humidity, air pressure, and wind speed, which were used to drive the model.

**N fertilization data:** The historical chemical fertilizer (1961–2010) and manure (1860–2014) application data for

croplands were derived from the datasets produced by Nishina et al. (2017) and Zhang et al. (2017), respectively.

The synthetic N fertilization dataset is mostly based on country-specific information from the Food and Agriculture Organization statistics (FAOSTAT) after filling data gaps (Nishina et al., 2017). Notably, the dataset provided application date and monthly input N fertilizer differentiated into $NH_4^+$ and $NO_3^-$ considering the seasonal crop calendars for the dominant crops in each grid (Sacks et al., 2010). The synthetic N application rates in 2011–2015 were assumed to be the same as that

for 2010. In addition, if the amount and type of N fertilizer from Nishina et al. (2017) failed to match the site information obtained from the literature, we utilized the site-specific amount of fertilizer application according to the published paper and the $NH_4^+$-N/ $NO_3^-$-N ratio provided by Nishina et al. (2017).

The manure N dataset (Zhang et al., 2017a) included the annual manure production and annual application, which were reconstructed using the dataset from the Global Livestock Impact Mapping System (GLIMS) in conjunction with country-

specific annual livestock populations and the gridded cropland distribution map for 1860–2014 obtained from HYDE 3.2 (Goldewijk et al., 2017). The manure N production and application rates in 2015 were assumed to be the same as those in 2014.



**N deposition data:** We extracted the annual N deposition data based on the global maps of atmospheric nitrogen deposition (1993) (Dentener, 2006; http://daac.ornl.gov/cgi-bin/dsviewer.pl?ds_id5830) supported by a three-dimensional global chemistry transport model (TM3) (Lelieveld and Dentener, 2000), which used N emission estimates (van Aardenne et al., 2001) and projection scenario data (Houghton, 1996; Nakicenovic et al. 2000).

**Vegetation:** For the model initialization, we generated vegetation cover data by overlaying the Global Land Cover Map for 2009 (GlobCover2009) based on Medium Resolution Imaging Spectrometer (MERIS) remote sensing data (http://due.esrin.esa.int/page_globcover.php) with the ecoregions framework from the World Wildlife Fund (WWF). Then, we generated a new category of global vegetation cover types that fitted the plant functional type of the model and relied on these land cover data. The annual cropland area from 1860 to 2015 was acquired from the History Database of the Global Environment, version 3.2 (HYDE 3.2), which has reconstructed time-dependent land use using historical population and allocation algorithms with weighting maps (Goldewijk et al., 2017). Cropland can be classified into rain-fed and irrigated land, both of which were further divided into rice, generic $C_3$ crops (except rice, e.g., wheat), and generic $C_4$ crops (e.g., maize) based on the global crop distribution maps (Monfreda et al., 2008).

**Soil data:** The global soil properties (soil texture and soil pH) and classification were obtained from the Food and Agriculture Organization/United Nations Educational, Scientific and Cultural Organization (FAO/UNESCO) Soil Map of the World (http://www.fao.org/geonetmork/srv/en/metadata.show?id514116) and the dataset provided by Batjes (2006), respectively. The soil C and C:N ratio data used for the model initialization were generated from a global soil dataset (IGBP-DIS; 2000).

**Topographic data:** We used a global digital elevation model (DEM) with an approximate spatial resolution of 1 km (GTOPO30) for the topography input (http://www.temis.nl/data/gtopo30.html).

**Atmospheric $CO_2$ concentration data:** The monthly atmospheric $CO_2$ concentration data for the simulation period from 1860 to 2015 was obtained from the National Oceanic and Atmospheric Administration (NOAA) GLOBALVIEW-$CO_2$ dataset derived from atmospheric and ice core measurements (www.esrl.noaa.gov).

### 3.2.3 Model Calibration and Validation

The daily $N_2O$ flux data for 39 sites were used for the model calibration, and the mean daily emission data for 69 other sites were used for the model validation. We estimated the model parameters, soil properties, and vegetation information from our input datasets, and used the agricultural practice's information obtained from the corresponding literature (Table 2), e.g., the amount of N input, to set up the model.

Before the model simulation and analysis, a spin-up period of about 300 years was conducted until the soil biogeochemical cycles and the compositions of the different C and N pools remained in equilibrium under stationary climate





conditions, which was the multiyear mean climate data.

For the model calibration, we used the daily climate data for each site to drive the model along with other site-specific

input information. The simulation started on January 1st, 1901, and ended on December 31st, 2015, with a daily time step. By

comparing the output N₂O flux data with the observed data, we adjusted the most sensitive parameter of the N₂O emissions

based on the sensitivity analysis in order to fit the best model performance via trial and error and statistical model performance

indicators. The index of agreement (*D*), the root mean square error (*RMSE*), and the coefficient of determination (*R²*) were

used to evaluate our model's performance, and the D-value and *RMSE* were calculated as follows:

$$D = 1 - \frac{\sum_{i=1}^{n}(S_i - O_i)^2}{\sum_{i=1}^{n}(|S_i - \overline{O}| + |O_i - \overline{O}|)^2},$$     (8)

$$RMSE = \sqrt{\frac{\sum_{i=1}^{n}(S_i - O_i)^2}{n}}.$$     (9)

Here, $S_i$ is the ith simulated result corresponding to the number of observations; $O_i$ is the ith observed value; and $\overline{O}$ is the

mean of the observed values during the experimental period. D varies between 0 and 1, and is excessively sensitive to extreme

values (Willmott, 1981). The model performance was considered to be perfect and unmeaningful when the D value was set to

1 and 0, respectively. The *RMSE* is the key value representing the difference between the simulated and observed values, and

is significantly affected by the data units.

Based on the calibration results and the fitting of the most sensitive parameter for the different sites, we used the continental

mean parameter for the model validation.

**4. Results**

**4.1 Sensitivity analysis**

The mean sensitivity index (*SI*) varied from -0.53 (EFF_NO2) to 1.37 (COE_dNO3) for the selected 13 parameters (Fig. 2). All

of the parameters had a nonunique effect on the N₂O emissions of the different sites. COE_dNO3, COE_NR, MUE_NO3, M_NO3, EFF_N2O,

COE_dNO2, and COE_dNO mostly had positive effects, while the remaining parameters either had negative effects (e.g., MUE_N2O

and EFF_NO2) or had no evident impact (e.g., AMAX) on the N₂O fluxes. The coefficient of the NO₃⁻ consumption rate (COE_dNO3)

was the most sensitive parameter in the current TRIPLEX-GHG model. The *SI* ranged from -0.61 to 5.39 (with a mean of 1.37)

for the current model input information. We also noticed that the *SIs* of the selected parameters were not consistent with the

different input information, especially for the variations in the amount of N fertilizer applied. The COE_dNO3 slightly increased

initially and then decreased as the N dose increased; and as the most sensitive parameter, it retained a large *SI* value (Fig. S2).

Overall, to simplify the parameter fitting processes and to evaluate the model's performance, we selected COE_dNO3 as the

fitting parameter, while we set the other parameters to their original constant values as the default (Table 1).

**4.2 Model calibration**





The calibration sites were categorized into six main regions according to their geographical distribution, including North

America (NA), Asia (AS), Europe (EU), Australia (AU), South America (SA), and Africa (AF). Generally speaking, the

model's performance was reasonably good in terms of the comparison of the site observations with the modeled results (Table

3).

### 4.2.1 North American sites

The data collected for the North American cropland sites were located in the US and Canada and represented the dominant

commercial crop species such as corn, wheat, barley, and tomatoes. Most of the measurements were collected over more than

two years. For the sites located in the great lakes region (NA-1 and NA-2), the modeled seasonal patterns of the $N_2O$ emission

were generally consistent with the measured data (Figs. 3a–b), but the estimated pulses had longer durations than the

observations (the model could not capture the detailed variations in the detected $N_2O$ fluxes), which resulted in low agreement

indices ($D$=0.65, $D$=0.56). For the studies carried out in the eastern Atlantic coastal region, the annual variation in the field

data from site NA-3 was reproduced well by the model (Fig. 3c), except for some underestimated peak values, which slightly

reduced the level of the model evaluation indices ($D$=0.69, $RMSE = 3.6$, $R = 0.57$). Furthermore, the modeled simulation results

were well matched for the scattered detected values of sites NA-4 and NA-5 (Figs. 3d–3e), with model agreement indices of

0.81. The model's results were also strongly correlated with the other collected observation data in the central (NA-6), southern

(NA-7) USA, and western coastal regions of the continent (NA-8, NA-9). The model performed well for the long-term

fertilized corn sites in Colorado (Fig. 3f; $D = 0.84$, $RMSE = 0.90$, $R = 0.73$). Nevertheless, the model's results showed relatively

low evaluation indices ($D = 0.59$, $RMSE = 4.09$, $R = 0.40$) due to underestimating the length of the intensive emission period

in July 2014 at site NA-7 (Fig. 3g). Moreover, its failure to capture the emission peaks in 2011/10 and 2012/10 for the California

tomato site slightly jeopardized the model's performance (Fig. 3h; $D = 0.61$, $RMSE = 0.87$, $R = 0.48$). As for site NA-9, the

general trends of the modeled $N_2O$ flux results were consistent with the observation data (Fig. 3i; $D = 0.75$, $RMSE = 0.92$, $R$

$= 0.84$).

### 4.2.2 Asia

Ten upland agricultural sites were selected in Asia (Table 2 and Fig. 4), including one in central India (AS-1), one in

Japan (AS-7), one in the Aral Sea Basin, Uzbekistan (AS-10), and several in China. All of the selected sites were characterized

by long-term cultivation histories and intense agricultural activities.

In general, the model captured the main variations in the observations and agreed well with all of the daily observations

for most of the sites, except for conventional cropland sites AS-1 and AS-2. The observed $N_2O$ variations in site AS-1 were

overestimated (2009/7, 2010/1, and 2010/4) and underestimated (2008/7 and 2010/4) compared with the simulated results,

leading to an agreement index of 0.69 (Fig. 4a). According to the observed $N_2O$ emission rates for site AS-2 reported by Guo

et al. (2013), certain points were being recorded as negative values without apparent regularity in the time series, while the





model was less robust in terms of capturing the occurrence of N$_2$O uptake, resulting in a low index of agreement (Fig. 4b, $D =$

0.50). In addition, the simulation exhibited reasonable N$_2$O flux variation patterns, especially the occurrence of emission pulses

induced by fertilization, comparable to those described by Zhou et al. (2019) (Fig. 4c) and Zhang et al. (2016) (Fig. 4d), while

the inaccurately estimated peak values suppressed the evaluation of the model's performance (AS-3, $D = 0.67$; AS-4, $D = 0.64$).

    For the other selected sites in Asia, the model results for sites AS-5, AS-6, AS-7, and AS-8 showed that simulated N$_2$O

fluxes agreed well with the observed fluxes under different agricultural practices, with model agreement indices of 0.86, 0.81,

0.78, and 0.76, respectively (Figs. 4e–h). Scattered observation points in a peanut site located in central-subtropical China

were also simulated by our model and the result showed a similar general pattern of N$_2$O flux with acceptable model

performance indices (Fig. 4i; $D = 0.65$, $R = 0.49$, $RMSE = 0.31$). For the long-term wheat cultivation site in Uzbekistan,

characterized by extremely high emission rates (>50 mg N m$^{-2}$ day$^{-1}$), the simulated N$_2$O emission rate matched the

observations well, except for one overestimated emission pulse in 2005/7 (Fig. 4j), which resulted in model performance

indicators of $D = 0.74$, $R = 0.60$, and $RMSE = 6.03$.

### 4.2.3 Europe

    Most of the wide-spread crop types were included in the calibration of the model simulation of the European cropland

sites, which were located in the mid-high latitude region. The simulated trends and magnitudes of N$_2$O were generally

consistent with the measured data for most of the sites, but some of them had relatively low agreement indices. Based on the

studies of Kavdir et al. (2008) and Senapati et al. (2016), the frequent failure of capturing the major emission pulses, such as

the one induced by fertilizer input in 2003/1 for EU-1 (Fig. 5a) and the one that occurred in 2013/6 for EU-2 (Fig. 5b),

accounted for the low agreement indices ($D = 0.51$ and 0.53, respectively). Moreover, the low evaluation indices of site EU-3

($D = 0.52$) are attributed to the estimation gap between the simulated and observed peak values and the duration time (e.g.,

overestimated emission peak in 2004/7) as well as the underestimation of the background emissions (Fig. 5c). The study carried

out by Hall et al. (2010) reported extremely high N$_2$O emission rates due to the application of large amounts of manure. The

model had a low agreement index because it underestimated the major peaks and the duration (Fig. 5d; $D = 0.61$).

    Additionally, the model simulation also revealed good agreement with the measured N$_2$O emission data for the other

European sites. For the site observations provided by Sosulski et al. (2015) and Baggs et al. (2003), the modeled emission rates

matched the observed scatter points reasonably well (Figs. 5e–f), with good agreement indices ($D = 0.75$ and 0.79, respectively).

As for sites EU-7 and EU-8, the modeled daily N$_2$O emission rates reflected the general trends of the N$_2$O emissions in response

to fertilization and irrigation practices well. However, the modeled results still mis-captured the minor emission pulses in

2009/1 at site EU-7 (Fig. 5f; $D = 0.77$, $RMSE = 1.46$, $R = 0.66$) and in 2007/8 at site EU-8 (Fig. 5g; $D = 0.87$, $RMSE = 0.23$,

$R = 0.75$). The model is sensitive to fertilizer application and produced well-simulated results comparing with the measured

data collected in Madrid (Fig. 5h; $D = 0.91$, $RMSE = 1.36$, $R = 0.88$).


### 4.2.4 Oceanic

Almost all of the cropland $N_2O$ studies carried out in Australia were located in the eastern coastal region, and only one rainfed continuous wheat site in western Australia was used in the model calibration. The low model evaluation indices of site AU-1 ($D$ = 0.47, $RMSE$ = 0.12, $R$ = 0.25) are probably associated with the failure to capture the emission peaks in 2006/1, 2007/4, and 2010/3 (Fig. 6a). For the other sites in eastern Australia, the general seasonal patterns of the simulated $N_2O$ emission were consistent with the observations. The model performed reasonably well for manure dominated site AU-2, and the overestimated peak value was responsible for the low agreement index (Fig. 6b; $D$ = 0.69). A lychee (*Litchi chinensis*) orchard site with a high sampling frequency was included, so we used the daily mean flux for the comparison. Notably, the PFT was considered to be subtropical forest for this site, and the model performed well (Fig. 6c; $D$ = 0.80, $R$ = 0.75) even though there was an obvious mis-capture of the emission peak in 2008/6. It should be noted that sugarcane was planted at site AU-5. Because the C properties of sugarcane differ significantly from those of grain crops (e.g. wheat), the PFT was set as shrub during the calibration. The modeled results of the sugarcane-based crop systems agreed well with the measured data (Fig. 6e; $D$ = 0.73, $RMSE$ = 0.65, $R$ = 0.55).

### 4.2.5 South America & Africa

Unfortunately, there are insufficient observations of cropland $N_2O$ emissions conducted in the agriculturally dominated regions of South America and Africa (Fig. 7 and Table 2). Typical agricultural ecosystems in these regions (sugarcane, wheat) were selected for the model calibration. Compared with the results of the two sites with short experimental periods in Africa (Figs. 7a–b), the simulated seasonal $N_2O$ variation agreed reasonably well with the one year of observations as is indicated by model performance indices (AF-1: $D$ = 0.92, $RMSE$ = 0.22, $R$ = 0.94; and AF-2: $D$ = 0.87, $RMSE$ = 0.65, $R$ = 0.93).

In South Africa, both cereal and economic crop sites were included. The model results were in good agreement with the measured $N_2O$ emission rates reported by Passianoto et al. (2003) even though the number of points were limited (Fig. 7c; $D$ = 0.93, $RMSE$ = 0.70, $R$ = 0.90). Moreover, the modeled results also illustrated that the $N_2O$ variation patterns for the model simulations and the observations are good agreement for the maize-wheat site SA-2, but the model mis-captured minor pulses, slightly reducing the evaluation index (Fig. 7d; $D$ = 0.81, $RMSE$ = 4.19, $R$ = 0.67). For the sugarcane site SA-3, the simulated results were generally well correlated with the measured $N_2O$ fluxes, which are highly regulated by the agricultural practices; however, the model failed to capture the consistent relatively high-level emission rates after fertilizer application (Fig. 7e; $D$ = 0.74, $RMSE$ = 1.25, $R$ = 0.65).

In summary, according to the calibration results, the trends and magnitudes of the simulated $N_2O$ flux were generally consistent with the measured field data.

### 4.3 Model validation

The model validation (Fig. 8) involved comparing the simulated and measured daily mean of the $N_2O$ emissions for all



of the validated sites, and the results are also presented in Table S2. During the validation, the simulated daily mean emission rates during the experimental periods ranged from 0.048 to 5.21 mg N $m^{-2}$ $day^{-1}$, and most of the values were less than 1 mg N $m^{-2}$ $day^{-1}$. The regression result was close to the 1:1 line, indicating that the modeled results are quite consistent with the

observed $N_2O$ emissions ($R^2 = 0.86$, $p<0.001$). However, the modeled results tend to slightly underestimate the $N_2O$ flux for the low observation values (<1 mgN $m^{-2}$ $day^{-1}$) and to overestimate for large observed $N_2O$ flux values (>1 mgN $m^{-2}$ $day^{-1}$). The model validation results further confirm that our model is capable of simulating the impacts of both climate and agricultural practices on $N_2O$ emissions across global cropland ecosystems.

**5. Discussion**

It is important to calibrate the process-based model using reasonable parameters in order to simulate complex biogeochemical processes better. Adjusting the most sensitive parameter is an efficient method of improving the model performance and has been widely used in model development and parameterization (Wang and Chen 2012; Zhang et al., 2017b; Zhu et al., 2014). As was stated in a previous study, Zhang et al. (2017) tested 23 parameters and found that the $COE_{NR}$, the

coefficient of the nitrification rate, controlled the $N_2O$ emission process. Meanwhile, our sensitivity analysis results revealed that the coefficient of the nitrate consumption rate, $COE_{dNO3}$, had the highest sensitivity level for the updated version of the TRIPLEX-GHG model. Such a divergence is probably due to the increased N input, especially $NO_3^-$, in cropland ecosystems compared with natural grasslands and forests. Denitrification strongly contributes to $N_2O$ production in agricultural ecosystems, which requires $NO_3^-$ as a substrate (Wang et al., 2018a). Since it is controlled by this parameter, the $NO_3^-$ consumption (from

$NO_3^-$ to $NO_2^-$) dominates the denitrification rate and thus the $N_2O$ production rate of N fertilized soil. Globally, the $COE_{dNO3}$ exhibited a large range of variation during the parameterization, which can partly be reconciled by the calibration method and the varying amounts of mineral N input. Because the $NO_3^-$ consumption rate for denitrification is difficult to measure directly, the limited field information strongly discourages the systematic adjustment of the $COE_{dNO3}$, and thus, the potential uncertainty of the parameter affected the model's performance.

Generally, the TRIPLEX-GHG model reproduces the $N_2O$ emissions well for a daily time step and various cropland ecosystems (e.g., wheat, maize, sugarcane, and cotton) on a global scale. The dominant characteristic of cropland $N_2O$ emission is the peaks associated with fertilization events, most of which were well simulated by our model and contributed to overall reasonable evaluation indices. Such advantages were derived from three features of our model. First, both the soil oxygen conditions and the soil water conditions were considered in the TRIPLEX-GHG model, i.e., represented by the size of the

anaerobic balloon and the water-filled pore space, respectively. Previous studies have highlighted that the soil $O_2$ status is the proximal, direct, and most decisive environmental trigger of $N_2O$ production (Song et al., 2019; Zhu et al., 2013; Khalil et al., 2004). However, the majority of process-based models only integrated the WFPS into the nitrification and denitrification



processes (e.g., Tian et al., 2010; Ito et al., 2018). It was reported that although the WFPS is a critical element containing information about the soil water and gaseous status, it still requires combination with other soil structural parameters in order to better predict the soil $O_2$ concentration, microbial respiration, and subsequent gas diffusion (Farquharson and Baldock, 2008; Song et al., 2019; Hall et al., 2013; Rabot et al., 2015). Second, a detailed description of the manure also contributed to the improved model performance because manure is a predominant soil organic carbon (SOC) source for croplands, which is not considered by empirical models and several of the process-based models (e.g., DAYCENT, VISIT). The SOC serves as a key energy and carbon source for microbial growth, nitrification, and denitrification (Snyder et al., 2009; Butterbach-Bahl et al., 2013). Field observations have shown that the application of manure either promotes or reduces $N_2O$ emissions probably because the added organic C compounds support microbial growth, but the increased SOC stimulates complete denitrification with the further reduction of $N_2O$ to $N_2$ (Zhou et al., 2017; Meijide et al., 2007). Therefore, in this study, the manure sourced C was recalculated using the manure N and C:N ratio, which significantly enhanced the simulation of the SOC. Last but not least, the TRIPLEX-GHG model included a reasonable microbial growth and death description, which strongly improved the accurate modeling of the nitrification process because the soil microbial conditions are one of the primary determinants of the soil nitrification rate at a global scale (Li et al., 2020).

However, there are still major discrepancies between the modeled and measured $N_2O$ fluxes, including underestimated peak values, failure to capture emission peaks, and underestimated background emissions. First, although the timing of the simulated major emission pluses was well simulated, the peak values of the emitted $N_2O$ fluxes were underestimated. The incomplete description of the processes involving the interaction between the soil pH and the external mineral N input is probably responsible for this phenomenon. The soil pH is one of the most important drivers of $N_2O$ production. Acidic soils are more sensitive to N input than alkaline soils, which probably enhances $N_2O$ production in croplands (Wang et al., 2018b; Morkved et al., 2007). Studies have shown that the pH values of agricultural soil tend to be significantly reduced by N deposition and N-fertilization at the global scale (Tian and Niu, 2015; Godsey et al., 2007; Guo et al., 2010). However, because the soil buffer capacity is difficult to quantify (Baron et al., 2014; Zhang et al., 2017b), the soil pH in our model was input information with a consistent pH value for each grid, and we neglected the effect of N input on soil pH such as the hydrolysis of urea (Tian and Niu, 2015; Wang et al., 2018b).

Next, the simulated results occasionally failed to capture several peaks in the observed $N_2O$ emission values. The mis-capture or underestimation of these peak values became evident in early spring when freeze–thaw events occurred (Figs. 4h and 5g). Freeze–thaw induced $N_2O$ emission pules constitute a major component of the annual total $N_2O$ emission at high latitudes (Wagner-Riddle et al., 2017; Kim et al., 2012) because increased soil temperature significantly promotes both soil physical mechanisms and microbial metabolism (Wolf et al., 2010; Wagner-Riddle et al., 2017). The former helps release the trace gases accumulated and trapped within the ice layer, and it simultaneously stimulates the formation of anaerobic conditions





(Teepe et al., 2004; Groffman et al., 2006). The latter triggers microbial driven nitrification and denitrification processes

(Sharma et al., 2006). The limited description of those processes, especially the simple empirical parameters and algorithms

we used for modeling snow-melting hydrology and nutrient release, are the primary error sources (Zhang et al., 2017b).

Furthermore, as for the underestimated background emissions, it is still a significant challenge for the process-based

model to accurately quantify background $N_2O$ emissions due to the following possible reasons. First, our simulations used

general crop classification ($C_3$, $C_4$, and rice) instead of detailed crop rotation information with different physiological

parameters (Ito et al., 2018; Monfreda et al., 2008; Saikawa et al., 2013). Field observations have revealed that different crop

types or species have diverse impacts on the $N_2O$ fluxes of cropland (Rochette et al., 2018; Philibert et al., 2013; Petersen et

al., 2006; Gelfand et al., 2016). For instance, legume species (e.g., soybean) have a stronger N fixation ability, which

contributes considerably to the N pools in cropland soil (Liu et al., 2010), and they effectively promote background $N_2O$

emission even without N fertilization compared with other cereal crops (Lenka et al., 2017; Sanchez and Minamisawa, 2019;

Yang and Cai, 2005). Second, in addition to climate conditions, the background emission rates from agricultural soils are also

associated with the amounts of residual N added in preceding years (Aliyu et al., 2018; Gu et al., 2009), and thus, the types of

residuals also have varying effects on the $N_2O$ emissions (Shan and Yan, 2013). Our model used the global mean ratio of the

returned residual N to the total plant biomass N for the simulation (Liu et al., 2010; Meng et al., 2005; Zhou et al., 2017)

because these agricultural practices are controlled by the individual farmers and vary greatly at the local and subregional scales,

without clear global distribution patterns such as those for soil and climate (Wang et al., 2018b). Third, the uncertainties in the

site history are also responsible for the inaccuracy of the modeled background emissions because the site history has a

tremendous effect on the soil properties, especially the SOC content (Gelfand et al., 2016). Previous studies have demonstrated

that agricultural practices, such as returning residues to the soil, tillage management, and fertilizer application, are important

drivers of the SOC (Liu et al., 2014; Jiang et al., 2018; Zhou et al., 2017), but their effects vary with the intensity of the

practices and the climatic conditions (Ogle et al., 2019; Snyder et al., 2009; van Kessel et al., 2013; Liu et al., 2014; Gattinger

et al., 2012). Unfortunately, only a few published papers have provided detailed historical land use and agricultural practice

information, which is a barrier to the accurate estimation of the local SOC and thus $N_2O$ emissions.

Last but not least, the other reasons for the discrepancies between the modeled results and the observations may be the

uncertainties in the field measurements and the driving data. For one, the lower sampling frequency of the fieldwork and the

short-lived $N_2O$ emission pluses are particularly difficult to captured with traditional manual chambers, especially after base

fertilizer application in the fallow season (Lammirato et al., 2018; Lognoul et al., 2019). Moreover, the model's accuracy also

relies on good quality data. A 0.5°×0.5° global scale daily climate input dataset was used for the model calibration and

validation, but it is unlikely that every site was provided with detailed meteorological information due to the relatively coarse

spatial resolution. The local climate may differ significantly from that of the grid input information (Wania et al., 2010).





Specifically, the precipitation information is less accurate compared with the other climate data, which could significantly

jeopardize the model's performance since the anaerobic balloon is a precipitation-induced process (Zhang et al., 2017b).

Furthermore, the soil properties are also difficult to be precisely replicate at the site level using a global soil dataset. Because

the soil texture served as a significant driver for the $N_2O$ emissions (Philibert et al., 2013; Gu et al., 2013), the mismatch of

soil information is also a major cause of the disagreement between the model simulations and observations.

In response to the uncertainties described above, further modeling is suggested to improve the detail of the descriptions

of the key processes, and better quality datasets need to be collected.

First, it is recommended that more detailed soil microbial activities be considered in order to better model the features of

the $N_2O$ emissions (Li et al., 2020). The nitrifier-denitrification process may account for up to 100% of the $N_2O$ emissions

from $NH_4^+$ in soils (Wrage et al., 2001; Wrage-Moennig et al., 2018), especially for $N_2O$ uptake, the occurrence of which has

been widely observed in peatlands, boreal forests, (Saikawa et al., 2013) and occasionally in cropland ecosystems (e.g., Fig.

4b). In addition, ammonia oxidation has been found to be a significant process for the development of $N_2O$ compared to

classical denitrification in extremely low-oxygen concentration soils (Zhu et al., 2013).

Next, in this study, only general PFTs were used for croplands without specifying crop types, for which the nutrient

requirements, maximum productivities, C:N ratios of different organs, and biomass allocation patterns differ significantly from

each other (Li et al., 2000; Shan and Yan, 2013), which significantly affects the N dynamics of cropland soils.

Furthermore, various agricultural management practices were not included in the current model. For example, different

techniques of tillage (e.g. conventional tillage, minimum tillage, tillage with different instruments), irrigation (e.g., flood

irrigation and drip irrigation), and fertilizer placement (e.g., top dressing and injection) can have diverse impacts on $N_2O$

emissions (Maris et al., 2015; Rochette, 2008). For instance, drip irrigation effectively promotes WFPS without surface runoff,

which induces significant $N_2O$ flux (Sanchez-Martin et al., 2008). Considering the advantages of field studies, model

performances can be effectively improved at the site level.

### 6. Conclusions

Our study represents a successful attempt to fully integrate general agricultural activities into the current TRIPLEX-GHG

framework for simulating global $N_2O$ emissions across cropland ecosystems. In this study, the $COE_{dNO3}$, which controls the

$NO_3^-$ consumption rate of the denitrification process, was found to be the most sensitive parameter. The key parameter was

calibrated using measured data for 39 global cropland sites, and we found that the improved TRIPLEX-GHG model was

capable of simulating the dynamics and magnitudes of $N_2O$ emissions from croplands at a daily time step, especially the

emitted peaks associated with fertilizer application. The model validation results further confirm that the modeled $N_2O$

emissions were highly correlated with the observed data. However, the model was unable to capture several detailed emission



characteristics, which jeopardized the model's performance. Further development of the TRIPLEX-GHG model could contribute to sustainable agricultural development, scientific modeling, and a better quantification of the global greenhouse gas budget under global change.

**Code and Data availability**   The source code and configuration for the current version of the TRIPLEX-GHG model employed for the simulations is available at https://doi.org/10.5281/zenodo.4679490. All the model input datasets can be obtained from the cited publications and websites. Model input information for calibration and validation were derived from corresponding papers. Notably, the model output dataset can be required by contacting the first author.

**Author contribution**   Changhui Peng and Qiuan Zhu designed the study. Hanxiong and Kerou developed the model code and performed all the simulations and model tests. Hanxiong prepared the manuscript with contributions from all co-authors.

**Competing interests**   The authors declare that they have no conflict of interest.

**Acknowledgements**   This study was funded by the Natural Sciences and Engineering Research Council of Canada Discovery Grant and the National Key R&D Program of China (2016YFC0500203). We thank LetPub (www.letpub.com) for its linguistic assistance during the preparation of this manuscript.





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





**Table.1 List of the major parameters for processes associated with N₂O production.**

| Parameters | Explanation | Values | Unit | References |
|---|---|---|---|---|
| $COE_{dNO3}$ | Coefficient for consumption rate of $NO_3^-$ | 0.05 | | (Li et al., 2000; Zhang et al., 2017a) |
| $COE_{NR}$ | Nitrification rate coefficient | 0.044 | | (Cai et al., 2014; Zhang et al., 2017a) |
| NMUEMAX | Growth coefficient for nitrifiers | 0.102 | $d^{-1}$ | (Li et al., 2000) |
| AMAX | Mortality coefficient for nitrifiers | 0.06 | $d^{-1}$ | (Li et al., 2000) |
| $MUE_{NO3}$ | Maximum growth rate of $NO_3^-$ denitrifiers | 0.67 | $h^{-1}$ | (Li et al., 2000) |
| $MUE_{NO2}$ | Maximum growth rate of $NO_2^-$ denitrifiers | 0.67 | $h^{-1}$ | (Li et al., 2000) |
| $MUE_{N2O}$ | Maximum growth rate of $N_2O$ denitrifiers | 0.47 | $h^{-1}$ | (Li et al., 2000) |
| $EFF_{NO3}$ | Efficiency parameter for $NO_3^-$ denitrifiers | 0.501 | $h^{-1}$ | (Li et al., 2000) |
| $EFF_{NO2}$ | Efficiency parameter for $NO_2^-$ Denitrifiers | 0.428 | $h^{-1}$ | (Li et al., 2000) |
| $EFF_{N2O}$ | Efficiency parameter for $N_2O$ denitrifiers | 0.075 | $h^{-1}$ | (Li et al., 2000) |
| $M_{NO3}$ | Maintenance coefficient on $NO_3^-$ | 0.09 | $h^{-1}$ | (Li et al., 2000) Leffelaar, and Wessel 1988, |
| $COE_{dNO2}$ | Coefficient for consumption rate of $NO_2^-$ | 1.0 | | (Norman et al., 2008; Zhang et al., 2017a) |
| $COE_{dNO}$ | Coefficient for consumption rate of NO | 1.0 | | (Norman et al., 2008; Zhang et al., 2017a) |






**Table 2. Information on the selected sites used for the model calibration.**

| ID | Sites | Lat. | Lon. | Experiment period | Crop type | Fertilization (kgN ha⁻¹ yr⁻¹) | return straw | irrigate | clay | sand | pH | SOC (%) | Soil C:N ratio | Mean N₂O flux (mgN m⁻² day⁻¹) | Method | Reference |
|---|---|---|---|---|---|---|---|---|---|---|---|---|---|---|---|---|
| NA-1 | Woodslee, Ontario, CA | 42.1 | -82.6 | 2003-2005 | corn | 150 | yes | no | 52.5 | 20 | 6.4 | 12.5 | 4.7 | 0.47 | closed chamber | (Drury et al., 2008) |
| NA-2 | Rosemount, MN, USA | 44.8 | -93.1 | 2008-2009 | corn | 146 | no | no | 23.0 | 22 | 6.2 | 2.8 | 9.4 | 0.25 | stainless steel chamber | (Venterea et al., 2011) |
| NA-3 | Marlboro, MD, USA | 39.4 | -77.3 | 2012-2014 | Tobacco; corn | 134 | yes | no | 9.3 | 79.6 | 6.2 | 0.8 | 9.1 | 1.04 | static flux chamber | (Chen et al., 2018) |
| NA-4 | Fredericton, NB, CA | 45.9 | -66.6 | 2008-2011 | potato | 193 | no | no | 11 | 49 | 6.2 | 1.9 | 14.8 | 0.18 | non-steady-state chamber | (Zebarth et al., 2012) |
| NA-5 | Que'bec City, CA | 46.8 | -71.4 | 2002-2003 | corn | 150 | no | no | 48.2 | 11.2 | 6.9 | 3.4 | 14.6 | 1.68 | non-steady-state chamber | (Rochette et al., 2008b) |
| NA-6 | Fort Collins, Colorado, USA | 40.7 | -105.0 | 2002-2005 | corn | 134 | yes | no | 33.4 | 40.2 | 7.7 | 1.3 | 8.4 | 0.30 | automatedgas chambers | (Mosier et al., 2006) |
| NA-7 | Baton Rouge, LA, USA | 30.4 | -91.2 | 2013-2014 | cotton | 112 | no | no | 20.5 | 34.7 | 6.2 | 6.6 | 9.9 | 4.40 | closed chamber ph chromatographs | (Tian et al., 2015) |
| NA-8 | Sacramento County, CA | 38.3 | -121.5 | 2010-2012 | grape | 38 | yes | yes | 23 | 50 | 6.4 | 11.2 | 10.7 | 1.15 | closed chamber | (VERHOEVEN et al., 2014) |
| NA-9 | British CA | 49.2 | - | 2005-2007 | corn | 150 | yes | no | 14 | 27 | 6.1 | | 13.6 | 1.04 | static flux | (Hunt et al., 2016) |





| ID | Location | Lat | Lon | Year | Crop | | | | | | | | | | | Method | Reference |
|---|---|---|---|---|---|---|---|---|---|---|---|---|---|---|---|---|---|
| AS-1 | New Delhi, Inida | 28.2 | 77.2 | 2008-2010 | wheat | 120 | no | no | 22 | 52 | 8.1 | 0.6 | 8.6 | 0.27 | closed-chamber | (Jain et al., 2016) |
| AS-2 | Gongzhuling, Jilin,China | 43.5 | 124.8 | 2010-2012 | maize | 230 | yes | no | 23 | 39 | 6.2 | 2.6 | 9.0 | 0.72 | static closed-chamber | (Guo et al., 2013) |
| AS-3 | Yanting, Sichuan, China | 31.3 | 105.5 | 2012-2015 | wheat; maize | 300 | yes | no | 19.6 | 30.1 | 8.1 | 1.2 | 8.3 | 0.80 | static chamber | (Zhou et al., 2019) |
| AS-4 | Nanjing, Jiangsu, China | 32.1 | 119.0 | 2013-2014 | vegetable | 420 | no | yes | 54.5 | 15.2 | 5.5 | 1.5 | 8.8 | 3.52 | static chamber | (Zhang et al., 2016) |
| AS-5 | Yuncheng, Shanxi, China | 34.9 | 110.7 | 2008-2010 | Cotton | 70 | no | yes | 37.6 | 16.6 | 8.7 | 1.0 | 7.1 | 0.79 | Automatic chamber | (Wang et al., 2013) |
| AS-6 | Fengqiu, Henan, China | 35.0 | 114.3 | 2002-2003 | maize; wheat | 150 | yes | no | 6 | 79 | 8.7 | 8.9 | 7.9 | 0.23 | close-chamber | (Meng et al., 2005) |
| AS-7 | Japan | 36.0 | 140.1 | 2006-2007 | Komatsuna | 120 | no | no | 21 | 32 | 5.9 | 10.7 | 8.4 | 0.10 | automated chamber | (Hayakawa et al., 2009) |
| AS-8 | Shandong, China | 36.9 | 117.9 | 2008-2009 | Maize; wheat | 600 | yes | yes | 17.1 | 16.8 | 8.3 | 1.8 | 7.9 | 1.10 | static chamber | (Cui et al., 2012) |
| AS-9 | Xianning, China | 29.9 | 114.3 | 2005-2007 | peanut | 120 | yes | no | 2.4 | 49.0 | 5.2 | 0.9 | 4.9 | 0.34 | static closed chamber | (Lin et al., 2012) |
| AS-10 | Khorezm Uzbekistan | 41.6 | 60.5 | 2005-2006 | Cotton; wheat | 250 | yes | yes | 14.6 | 42.6 | 6.9 | 0.6 | 3.1 | 2.14 | closed chamber | (Scheer et al., 2008) |
| EU-1 | Potsdam Bornim, Germany | 52.4 | 13.0 | 2003-2005 | rape | 150 | no | no | 4 | 87.5 | 6.0 | 0.9 | 14.0 | 1.08 | static chamber | (Kavdir et al., 2008) |
| EU-2 | Lusignan, France | 46.4 | 0.1 | 2011-2014 | corn; wheat | 125 | no | yes | 17.6 | 13.2 | 6.4 | 13.5 | 10.6 | 0.34 | Automatic chamber | (Senapati et al., 2016) |





| | | | | | | | | | | | | | | | |
|---|---|---|---|---|---|---|---|---|---|---|---|---|---|---|---|
| EU-3 | St. Petersburg, Russia | 59.6 | 30.1 | 2003-2005 | potato | 120 | yes | no | 25.5 | 18 | 5.8 | 1.5 | 8.8 | 1.22 | closed chamber | (Buchkina et al., 2010) |
| EU-4 | BetDagan, Israel | 32.0 | 34.8 | 2006-2007 | cotton | 240 | yes | yes | 17.5 | 80 | 7.3 | | 10.3 | 9.42 | PVC sample chamber | (Heller et al.,2010) |
| EU-5 | Skierniewice, Poland | 52.6 | 20.3 | 2012-2013 | barley | 45 | yes | no | 7 | 87 | 6.6 | 11.0 | 11.1 | 0.44 | closed chamber | (Sosulski et al., 2015) |
| EU-6 | Wye Estate, UK | 51.9 | 1.0 | 1999-2001 | Wheat; rye | 200 | yes | no | 15 | 17 | 5.8 | 1.9 | 8.6 | 6.20 | closed chamber | (Baggs et al., 2003) |
| EU-7 | Stuttgart, Germany | 48.7 | 9.2 | 2008-2010 | vegetable | 401 | no | no | 30 | 2 | 5.5 | 1.8 | 8.0 | 1.85 | PVC-chamber | (Pfab et al., 2012) |
| EU-8 | Naples, Italy | 40.6 | 15.0 | 2007-2008 | maize | 130 | no | no | 32.9 | 47 | 7.5 | 0.8 | 8.4 | 0.10 | automated closed static chambers | (Forte et al., 2017) |
| EU-9 | Madrid, Spain | 40.5 | -3.3 | 2009-2012 | Maize; barley | 250 | yes | yes | 28 | 55 | 7.9 | 0.8 | 8.1 | 0.77 | closed chamber | (Abalos et al., 2013; Sanz-Cobena et al., 2012) |
| AU-1 | Cunderdin, Australia | -31.6 | 117.2 | 2005-2007 | wheat | 100;75 | yes | no | 18.6 | 77.0 | 6.0 | 0.4 | 10.0 | 0.032 | automated gas chambers | (Li et al., 2012) |
| AU-2 | Mackay, Queensland, Australia | -21.1 | 149.0 | 2006–2007 | Sugarcane | 150 | no | no | 33 | 38.5 | 4.7 | 1.7 | 9.4 | 1.61 | Automatic chambers | (Denmead et al., 2010) |
| AU-3 | Brisbane, Australia | -26.0 | 152.0 | 2007-2009 | lychee orchard | 256 | yes | no | 26 | 37 | 6.0 | 2.7 | 10.1 | 1.22 | automatic chambers | (Rowlings et al., 2013) |
| AU-4 | Queensland, Australia | -27.5 | 151.8 | 2009-2011 | Cotton; wheat | 200 | no | yes | 76 | 7 | 7.2 | 1.6 | 11.9 | 0.46 | automated chamber | (Scheer et al., 2012; Scheer et al., 2013; Scheer et al., 2016) |
| AU-5 | Queensland, Australia | -28.2 | 152.1 | 2006-2009 | wheat | 90 | yes | no | 65 | 11 | 6.9 | 2.0 | 9.7 | 0.83 | automatic gas sampling | (Wang et al., 2011) |




| | | | | | | | | | | | | | | | |
|---|---|---|---|---|---|---|---|---|---|---|---|---|---|---|---|
| d, Australia | | | | wheat | | | | | | | | | | chamber | Scheer et al., 2013; Scheer et al., 2016) |
| AU-5 Queensland, Australia | -28.2 | 152.1 | 2006-2009 | wheat | 90 | yes | no | 65 | 11 | 6.9 | 2.0 | 9.7 | 0.83 | automatic gas sampling chambers | (Wang et al., 2011) |
| AU-6 Wagga Wagga, Australia | -35.4 | 147.5 | 1993-1994 | ryegrass | 200 | no | no | 15.5 | 74.5 | 5.5 | 8.1 | 9.8 | 0.087 | automatic static chamber | (Galbally et al., 2010) |
| AF-1 Kaptumo, Kenya | 0.12 | 35.5 | 2013-2014 | vegetable | 110 | yes | no | 27.8 | 62.3 | 6.0 | 4.1 | 12.4 | 0.25 | static chamber | (Rosenstock et al., 2016) |
| AF-2 Kenya | -0.31 | 35.4 | 2015-2016 | tea | 150 | no | no | 59 | 20 | 3.9 | | 12.4 | 0.34 | static chamber | (Wanyama et al., 2018) |
| SA-1 Ariquemes, Rondnia State, Brazil | -10.5 | -52.5 | 2001-2002 | B.brizantha | 42 | yes | no | 23.5 | 71 | 5.3 | 6.0 | 9.6 | 0.89 | recirculating chamber method | (Passianoto et al., 2003) |
| SA-2 Santa Maria, Brazil | -29.7 | -53.7 | 2010-2011 | Maize and wheat | 125 | no | yes | 19.2 | 44.3 | 5.9 | | 11.1 | 1.66 | non-steady-state chambers | (Aita et al., 2015) |
| SA-3 Campinas, Brazil | -22.9 | -47.1 | 2011-2013 | Sugarcane | 120 | yes | no | 41 | 41.5 | 5.6 | 5.8 | 10.8 | 0.68 | PVC static chambers | (Soares et al., 2015) |





**Table 3. List of calibrated values for COE$_{dNO3}$ and the model performance indices for calibrated sites**


| ID | latitude | longitude | COE$_{dNO3}$ | $D$-value | $RMSE$ | $R$ | counts |
|----|----------|-----------|--------------|-----------|--------|-----|--------|
| NA-1 | 42.08 | -82.57 | 0.05 | 0.65 | 1.16 | 0.44 | 53 |
| NA-2 | 44.75 | -93.06 | 0.01 | 0.56 | 1.09 | 0.35 | 106 |
| NA-3 | 39.43 | -77.30 | 0.03 | 0.69 | 3.60 | 0.57 | 157 |
| NA-4 | 45.92 | -66.60 | 0.04 | 0.81 | 0.92 | 0.84 | 50 |
| NA-5 | 46.80 | -71.38 | 0.03 | 0.81 | 3.66 | 0.68 | 53 |
| NA-6 | 40.65 | -104.98 | 0.03 | 0.84 | 0.90 | 0.73 | 123 |
| NA-7 | 30.35 | -91.17 | 0.04 | 0.59 | 4.09 | 0.40 | 43 |
| NA-8 | 38.30 | -121.47 | 0.029 | 0.61 | 1.87 | 0.48 | 135 |
| NA-9 | 49.24 | -121.76 | 0.01 | 0.75 | 0.92 | 0.84 | 57 |
| AS-1 | 28.23 | 77.20 | 0.02 | 0.69 | 0.54 | 0.48 | 123 |
| AS-2 | 43.50 | 124.80 | 0.025 | 0.50 | 1.73 | 0.25 | 170 |
| AS-3 | 31.26 | 105.49 | 0.01 | 0.67 | 1.35 | 0.51 | 149 |
| AS-4 | 32.06 | 118.96 | 0.03 | 0.64 | 2.12 | 0.56 | 73 |
| AS-5 | 34.93 | 110.71 | 0.03 | 0.86 | 0.82 | 0.83 | 373 |
| AS-6 | 35.00 | 114.34 | 0.025 | 0.81 | 0.86 | 0.69 | 76 |
| AS-7 | 36.05 | 140.08 | 0.025 | 0.78 | 0.33 | 0.63 | 284 |
| AS-8 | 36.90 | 117.90 | 0.01 | 0.76 | 2.01 | 0.58 | 133 |
| AS-9 | 29.88 | 114.28 | 0.01 | 0.65 | 0.31 | 0.49 | 41 |
| AS-10 | 41.58 | 60.52 | 0.025 | 0.74 | 6.03 | 0.60 | 93 |
| EU-1 | 52.44 | 13.01 | 0.02 | 0.54 | 2.57 | 0.29 | 361 |
| EU-2 | 46.42 | 0.12 | 0.04 | 0.51 | 0.72 | 0.32 | 579 |
| EU-3 | 59.57 | 30.13 | 0.02 | 0.52 | 1.31 | 0.33 | 107 |
| EU-4 | 31.98 | 34.84 | 0.02 | 0.61 | 30.19 | 0.54 | 76 |
| EU-5 | 52.60 | 20.27 | 0.01 | 0.75 | 0.30 | 0.62 | 57 |
| EU-6 | 51.18 | 0.95 | 0.05 | 0.79 | 4.37 | 0.76 | 45 |
| EU-7 | 48.72 | 9.19 | 0.025 | 0.77 | 1.46 | 0.66 | 101 |
| EU-8 | 40.62 | 14.97 | 0.04 | 0.87 | 0.23 | 0.76 | 142 |
| EU-9 | 40.53 | -3.28 | 0.04 | 0.91 | 1.36 | 0.88 | 64 |
| AU-1 | -31.6 | 117.22 | 0.015 | 0.47 | 0.12 | 0.25 | 226 |
| AU-2 | -21.08 | 148.99 | 0.028 | 0.69 | 5.15 | 0.52 | 130 |
| AU-3 | -26.00 | 152.00 | 0.025 | 0.80 | 2.00 | 0.75 | 536 |
| AU-4 | -27.52 | 151.78 | 0.023 | 0.47 | 1.27 | 0.27 | 294 |
| AU-5 | -28.2 | 152.10 | 0.01 | 0.72 | 0.65 | 0.56 | 136 |
| AU-6 | -35.38 | 147.50 | 0.01 | 0.46 | 0.20 | 0.35 | 69 |
| AF-1 | 0.10 | 35.48 | 0.01 | 0.92 | 0.23 | 0.94 | 52 |
| AF-2 | -0.31 | 35.39 | 0.012 | 0.87 | 0.65 | 0.93 | 75 |
| SA-1 | -10.50 | -52.50 | 0.05 | 0.93 | 0.70 | 0.90 | 40 |
| SA-2 | -29.72 | -53.72 | 0.049 | 0.81 | 4.19 | 0.67 | 60 |
| SA-3 | -22.87 | -47.07 | 0.035 | 0.74 | 1.25 | 0.65 | 98 |



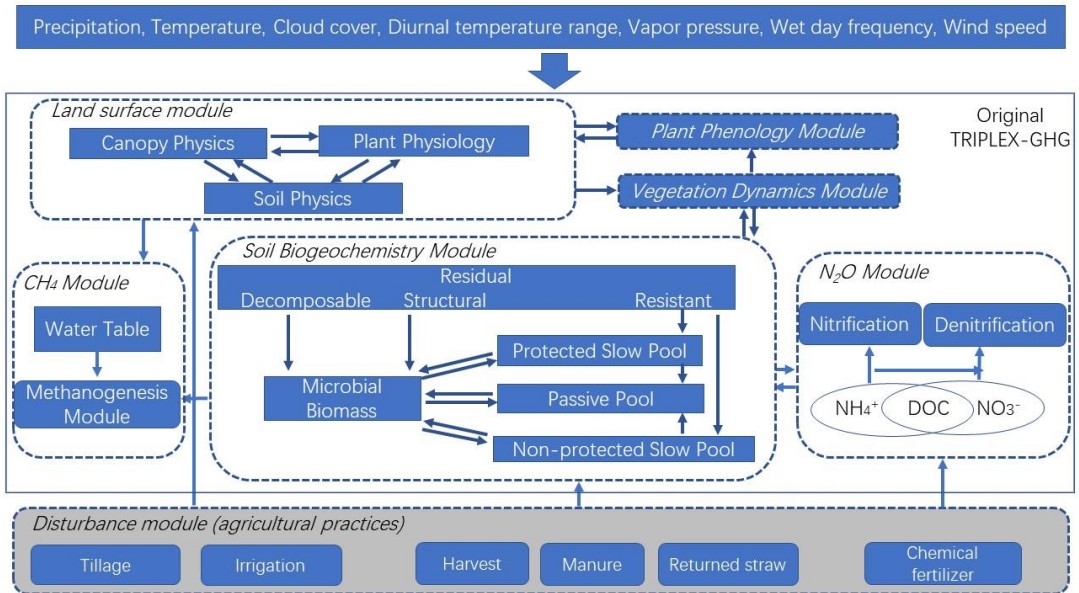

**Fig. 1 Model's structural concept and integration of agricultural practices into the TRIPLEX-GHG (revised from Zhang et al. (2017)). The rectangular insert with the light grey background represents the different agricultural practices and how they interact with the other submodules (e.g., the land surface module).**


.



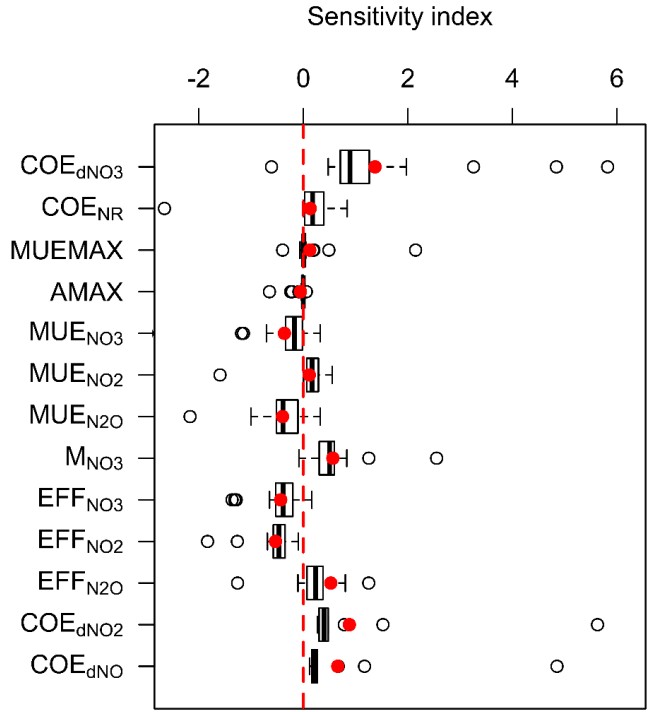

**Fig. 2 Sensitivity analysis of the different parameters. The closed red dots show the mean sensitivity index value of the parameters.**

**The outliers are shown as open dots.**



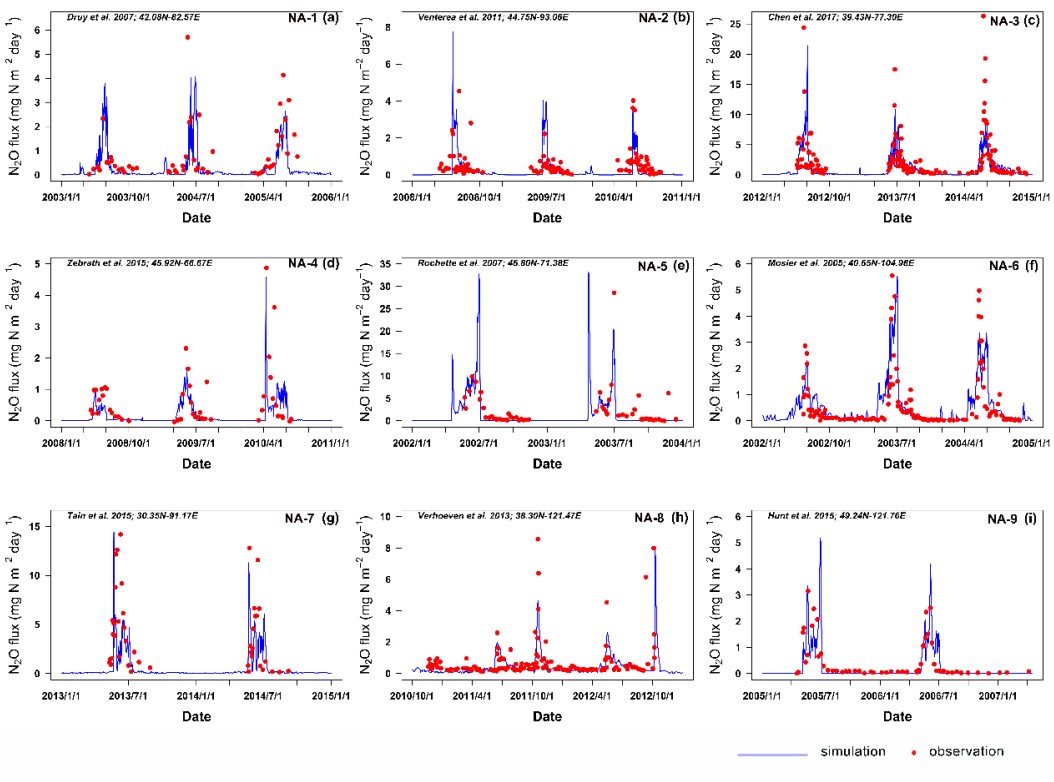

**Figure 3. Comparison of the modeled and observed N₂O emissions from the cropland sites located in North America.**







**Figure 4. Comparison of the modeled and observed N₂O emissions from the cropland sites located in Asia.**

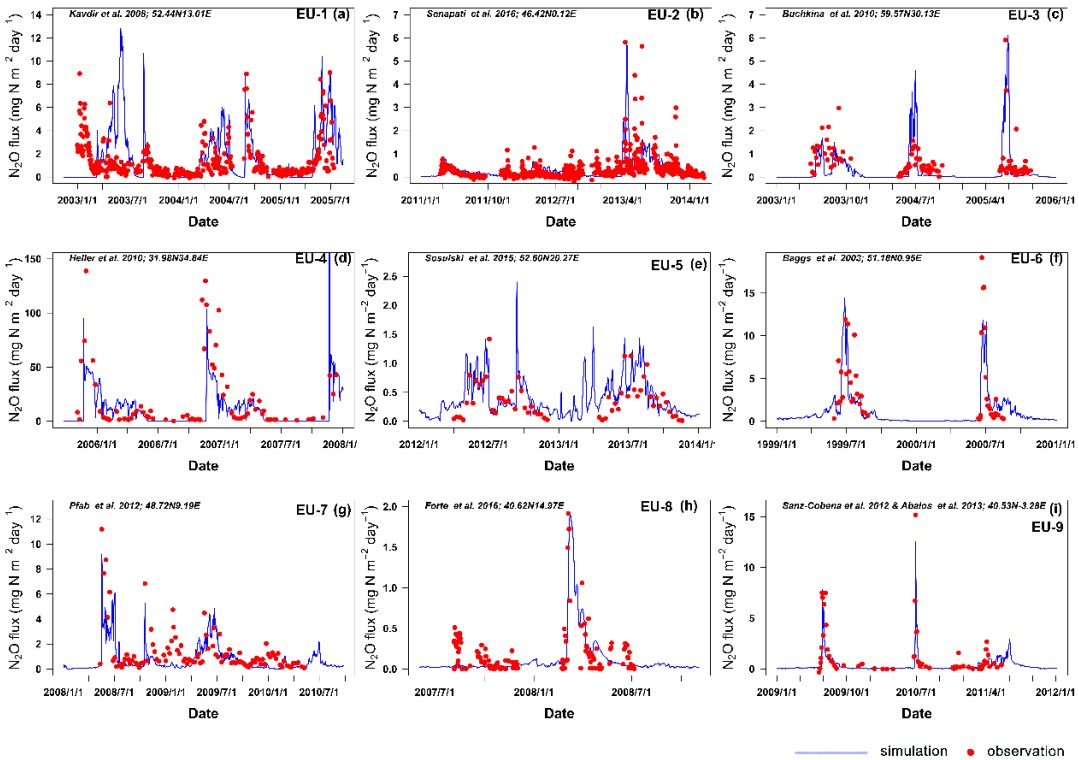

**Figure 5. Comparison of the modeled and observed N₂O emissions from the cropland sites located in Europe.**




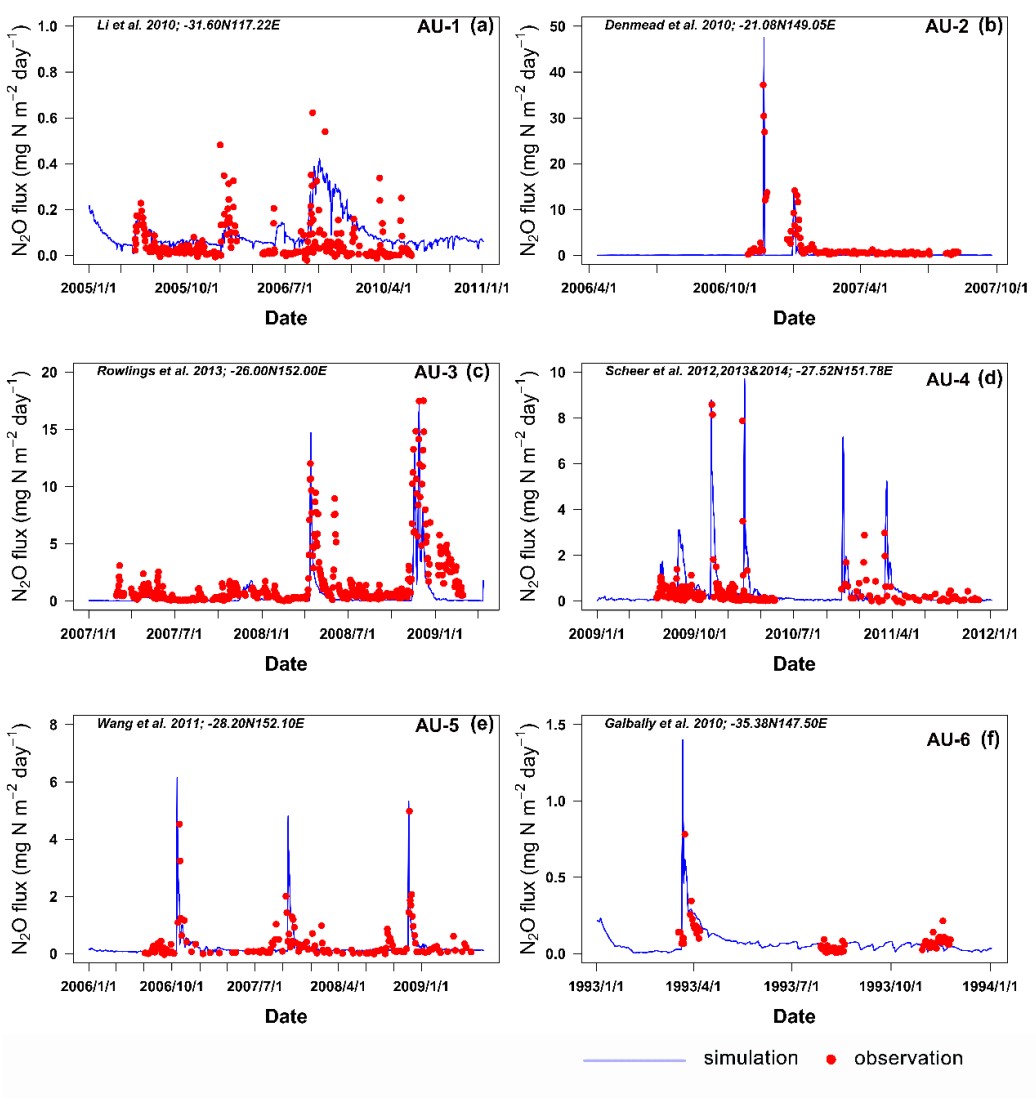

**Figure 6. Comparison of the modeled and observed N₂O emissions from the cropland sites located in Australia.**



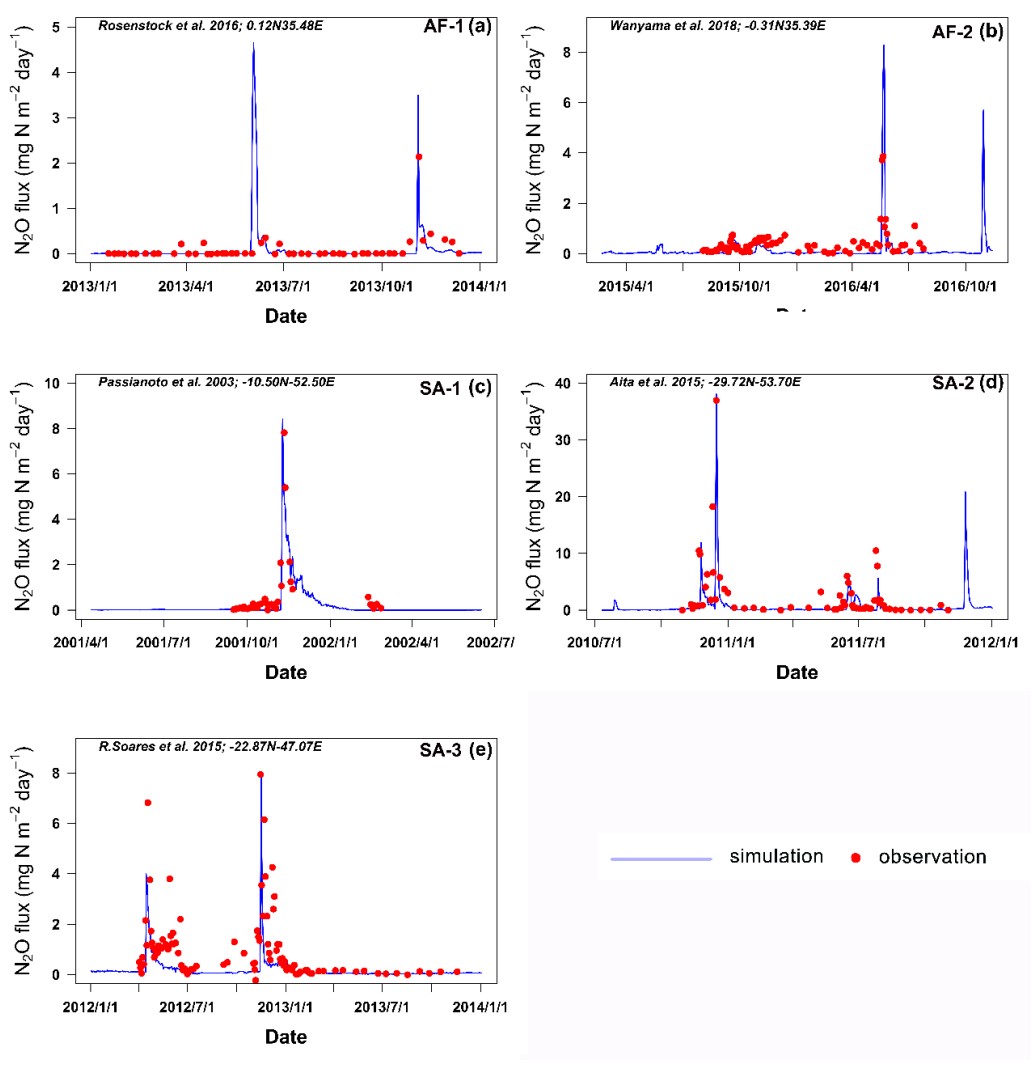


**Figure 7. Comparison of the modeled and observed N₂O emissions from the cropland sites located in Africa and South America.**





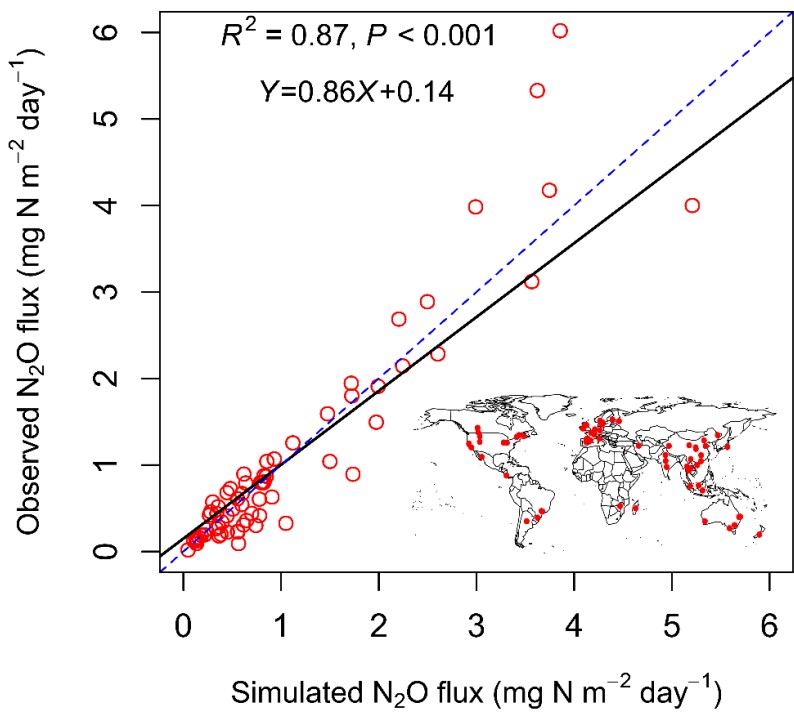

**Fig. 8. Comparison of the measured and modeled N$_2$O emissions from the validation sites (open red dots) and their global distribution (closed red dots on map).**