# Peer review of "Integrating Agricultural Practices into the TRIPLEX-GHG Model v2.0 for Simulating Global Cropland Nitrous Oxide Emissions: Model Development and Evaluation"

_Geoscientific Model Development, 2021_

## Author Comment (AC1)

Our point-by-point responses are provided below. The *referees' comments are italicized*, our answers are in blue and the **texts from the manuscript is highlighted in bold** for the Editor's easy reference. Some key issue we want to address is underlined:

*Song et al. developed a new version of the process-based TRIPLEX-GHG model to estimate $N_2O$ emissions from croplands by coupling major agricultural activities. The authors state that they found that the coefficient of the $NO_3^-$ consumption rate for denitrification was the most sensitive parameter based on their sensitive analysis result. I commend the authors for their effort to improve global $N_2O$ emissions from croplands as it is essential but I have some major issues with the paper that I believe need to be addressed before it can be published.*

**RE:** Thank you so much for your positive comments and feedback. We fully considered and addressed your suggestions and questions and made a point-by-point response below.

*First, the authors simulate daily $N_2O$ emissions and compare them to observational data. However, most of the measurements are often taken once a day, neglecting variations within a day, and so they are not representative for daily emission estimates. I am unsure how the authors have quantified daily estimates from existing literature. Also, there are different flux calculation schemes and for example, Venterea et al. (2020) illustrate a gold standard approach for calculating $N_2O$ flux. I wonder how many of the studies cited follow this approach and how these uncertainties in the observational data are taken into account.*

**RE:** Thanks for your question and suggestion about the measurement, method to estimate daily emission rate as well as the reliability of our model.

First, it is very true that most of the field measurements that we used for model testing took one gas sample a day as presented by the Table 2 column 'Method' which listed the ways to measure the $N_2O$ flux. Knowledge of the diurnal fluctuations in $N_2O$ flux has been used to choose a sampling time that maximizes the accuracy of $N_2O$ flux estimates, thereby reducing the sampling frequency required, but results from previous studies are inconsistent (Francis Clar and Anex, 2020). A general agreement for most of the studies is that the "Preferred Measuring Times" (PMTs) are between 10:00 and 12:00 AM, which means measuring the $N_2O$ fluxes at this time well represent the daily emission rate (Francis Clar and Anex, 2020; Reeves and Wang, 2015; Ferrari Machado et al., 2019). However, this measurement methods probably bring in some uncertainties during the emission hot moment (emission pulses after fertilization mostly) when high frequency of the measurement should be taken to ensure the reliability of reported flux data (Francis Clar and Anex, 2020).

In this study, we cited and utilized 107 observed sites for model calibration and validation and found that most of the chamber-based studies took the gas samples in the mid-morning which were in line with the recommended period (although some of them did not provide such information). In addition, consistent with the statement above, the variance of the measured emission data was larger during emission hot time (emission pulses) which means a larger uncertainty for this data. Fortunately, more frequent measurements were taken by most of the calibrated sites after fertilization and growing seasons (e.g., Fig. 4c, Fig. 4g and Fig. 6c). So, we believe those published papers provided relative reasonable reliability of the flux data, thus the reliability of the calibrated results of the model. The daily flux data we used from the literatures were captured and obtained with the software GetData Graph Digitizer. At least 40% of the calibrated sites did not include the

variance of the daily fluxes in the figures (e.g., Mosier et al. 2006), it was extremely difficult to obtain all of the measured data to show whether or not the modeled daily flux was within the range of the standard deviation or confidence interval. To our knowledge, it is common practices for all existing process-based model assessments that reported the comparison of $N_2O$ emission rate with daily time step to use the reported mean fluxes in order to evaluate the model performances (Senapati et al., 2016; He et al., 2020; Zhang et al., 2017).

As for the flux calculation schemes, we thank you for your information so that we can look into this problem. Most of the field observation did not pay great attention to the description of the flux calculation schemes by just saying that '$N_2O$ and CO2 fluxes were calculated from the slope of the linear temporal change in the concentrations of the chambers' atmosphere.' which is of majority (Pfab et al., 2012) or 'Fluxes were calculated from the linear or nonlinear increase in concentration (selected according to the emission pattern) in the chamber.' (Mosier et al., 2006). A small number of studies used nonlinear fitting only (Cui et al., 2012). Therefore, to sum up, for the sites included to test our model, only limited studies considered the 'gold' standard provided by Venterea et al. (2020) at least they did not claim so. This probably resulted in uncertainties of the measured flux data.

Anyway, we thank you for pointing out these questions which are very critical when judging the reliability of the model because of the potential uncertainties for the data we used to test the model. Therefore, we summarized, improved and restated the answers about the uncertainties associated with the measurement frequency and flux calculate schemes in the discussion section (please see page 22, lines 634-645).

**"For example, daily $N_2O$ flux data was used to calibrate the model while the lower sampling frequency of the fieldwork (e.g., once a day) probably failed to represent the daily $N_2O$ emission since the strong fluctuation within a day as suggested by micrometeorological methods (Lammirato et al., 2018; Lognoul et al., 2019; Jones et al., 2011). This uncertainty became even more evident during high emission rates periods (e.g., short-lived $N_2O$ emission pulses after base fertilizer application in the fallow season), casting shadow to the estimation of cumulative emissions (Francis Clar and Anex, 2020). In the meantime, the calculated daily $N_2O$ flux data used for model testing should also be questioned because most of the field observations used linear regression which had large uncertainties compared with other flux calculation schemes (Venterea et al., 2020). Therefore, flux measurements with high temporal resolution as well as more frequent sampling were required to reduce the uncertainties of measure $N_2O$ flux data to ensure a more reliable estimated cumulative emissions for models (Giltrap et al., 2020)."**

*Second, the authors write down equations in the paper without explaining the units and some of the assumptions are not well explained. For example, the authors state that $COE_{NO3}$ was set to 4.0 according to the model test (L. 160) but it is unclear what kind of test was conducted.*
**RE:** We are sorry for the missing units and descriptions of the assumptions. We added up the unit of associated variables in the revised manuscript and supplement materials (e.g., on page 7, line 154-156).

As for the parameter $COE_{NO3}$ is less important parameter comparing with $COE_{dNO3}$ which we

used to calibrate the model so that the detailed procedure to decide of the value $COE_{NO3}$ was not presented. We apologize for missing this information and the test was just simply comparing the estimated mean annual $N_2O$ emission levels with site information (please see page 7, line 154-157 and Table S2).

We did not provide a detailed comparison of soil $NO_3^-$ and $NH_4^+$ concentration between model output and reported data because the unit of simulated soil mineral N is kg N ha$^{-1}$ of our model instead of mg N kg$^{-1}$ dry soil which is mostly used and reported in the literatures. The unit transfer requires soil bulk density (g m$^{-3}$) and the depth of plow layer (m) but these information can not be provided by current model. You may ask why not use the published information (i.e., bulk density) to find a possible close answer? It can be explained with 3 reasons. First, although published papers provide the information of bulk density while there is still discrepancy between the soil dataset used by model and the reality. Next, the soil properties are varying with soil layers, so that we can not be sure of how many layers we are supposed to use for calculation. Finally, the plow layer depth was not reported although the commonly used size is 0.3m.

$N_2O$ flux is the target and the most interested variables for current study. Previous large scale process-based model did not chose to report the simulated soil mineral N variation with multiple reasons (Thorburn et al., 2010; Tian et al., 2010; Ito et al., 2018; Zhang et al., 2017). One possible reason is that the excessively high atmospheric N deposition rates probably results in overestimation of soil $NH_4^+$ concentrations.

Zhang et al. (2017) showed a trend of slightly overestimation of the $N_2O$ flux from natural grassland. This overestimation probably derived from the simulation of N plant uptake because TRIPLEX-GHG model v1.0 assumed that plant takes $NH_4^+$ first to satisfy the N demand and uses $NO_3^-$ until the soil $NH_4^+$ is used up (Kucharik et al., 2000; Foley et al., 1996; Zhang et al., 2017). Such design is un-realistic in terms of mechanism and also leads to excessive soil $NO_3^-$ concentration (Chalk and Smith, 2021; Daryanto et al., 2018). In general, $NO_3^-$ is usually more available for plants, owing to its higher mobility which leads to more rapid diffusion to root and easier access to plant as mass flow and diffusion is main pathway for N uptake (Daryanto et al., 2018). Theoretically, the reduction of soil $NO_3^-$ result in lower $N_2O$ emission and we hope that by comparing the modelled and reported $N_2O$ emissions, we can indirectly prove the effectiveness of current improved model design in terms of soil mineral N level. We also should highlight that due to the substantial external N input to cropland soil, the soil mineral N level exceed the crop N demands and result in large $N_2O$ emission (Shcherbak et al., 2014). Therefore, the effect of the $COE_{NO3}$ on cropland $N_2O$ flux is minor compared with that of natural soil. We hope you can understand this and the data was added in the supplementary material to show the effectiveness of the value of $COE_{NO3}$ (Table S2). We also changed the sentence in the manuscript as '**In cropland soil, $NO_3^-$-N is more easily absorbed by roots due to higher concentration and mobility (Malhi et al., 1988; Kronzucker et al., 1997; Chalk and Smith, 2021; Daryanto et al., 2018)**' (please see line 148-149). In the future, we will keep improving the model with more precise results of the variation of soil mineral N concentrations.

*Third, the authors state that the $NO_3^-$ consumption rate for denitrification was the most sensitive parameter based on their sensitive analysis result but it is also written that the authors selected the coefficient of the $NO_3^-$ consumption rate ($COE_{dNO3}$) as the fitting parameter to simplify the parameter fitting processes (L. 301). It is unclear to me how this variable was selected as the fitting*

*parameter and if it can really be considered as sensitivity analysis if all the other parameters were simply set to the original constant value.*

**RE:** Thanks for your question and we are sorry for the limited explanation of testing the sensitivity of the parameters.

First, we should clarify that it is supposed to be named 'sensitivity analysis of the model parameters' instead of 'sensitivity analysis' which is kind of misleading. We changed the subtitle 'sensitivity analysis' to '**3.1.2 sensitivity analysis of model parameters**' (please see line 254). We also added a sensitivity experiment for the new integrated processes (i.e., fertilizer application, irrigation etc.) of the model to highlight the model improvement (please see page 9-10, line227-251).

Next, we should highlight that it was '**The coefficient of the $NO_3^-$ consumption rate for denitrification ($COE_{dNO3}$) was identified to be the most sensitive parameter based on sensitivity analysis of model parameters**' (see line 16-17 of the revised version). This statement was constant through this paper. One exception probably was in the Discussion section of the previous version of the manuscript that '$NO_3$ consumption rate for denitrification was the most sensitive **processes**'. $COE_{dNO3}$ was a parameter to constrain the process which is widely used technique for modelling (e.g., DNDC,DAYCENT etc.) (Li et al., 2000; Tian et al., 2018; Ito et al., 2018).

In our study, sensitivity analysis of the response of $N_2O$ emissions to key model parameters are required to identify the most sensitive parameter or parameters before further model calibration and validation (as the fitting parameter). During the sensitivity analysis of model parameter, we changed the value of one parameter each time while keeping others as default value to compare the relative changes of the $N_2O$ emissions by saying '**We changed one parameter at a time, while holding the others fixed at default value to evaluate the response rate of the model output (i.e., in this case $N_2O$ emission) to the changed parameter**' (please see line 260-262). Because a previous study of TRIPLEX-GHG model v1.0 has conducted sensitivity analysis of parameters and only one variable, $N_2O$ emission, were focused, a priori assumption can be made about the linearity, monotonicity, or additivity of the model response to parameter changes. Therefore, the common practice 'changing one parameter at a time' (as described in our study, line 264-266) is applicable (Pappas et al., 2013; Ogejo et al., 2010).

During the calibration, we changed the value of the selected, most sensitive parameter (after sensitivity analysis of parameters), to fit the modelled and measured daily $N_2O$ flux data for each site. In the meantime, other parameters were set to original value (please see line 340-343), which is for calibration not for sensitivity analysis of model parameter. We further addressed this to prevent possible misunderstanding by saying '**For model calibration, we adjusted value of the most sensitive parameter of the $N_2O$ emissions (obtained from sensitivity analysis of parameters) in order to fit the best model performance by comparing the output of daily $N_2O$ flux data with the observed data obtained from published papers…**' in the method section (please see line 340-342).

To prevent possible misunderstanding, we also revised this sentence as "**Overall, to simplify the parameter fitting processes and to evaluate the model's performance, we selected the most sensitive parameter of the model, $COE_{dNO3}$, as the fitting parameter for model calibration, while we set the other parameters to their original constant values as the default during model calibration** (Table 1)." (please see line 394-396). Thanks for pointing out this.

*I find that there is a value to the paper but without the above issues being addressed, it is hard for me to recommend publication in GMD. I think more explanation of the sensitivity analysis itself is also essential.*

**RE:** Thank you again for your comments and suggestions. We revised the description of the integrated process and provided additional explanation of the sensitivity experiment to show the impact of the new incorporated agricultural practices on $N_2O$ as section 4.1. We also revised the section of sensitivity analysis of parameter to prevent possible misunderstanding. Further comparison between the modeled and observed Emission Factors (EFs) were conducted to further confirm the relative reasonable mechanism of the model in response to external N inputs. Hopefully you can satisfy with revised version of this manuscript.

*Minor comments:*

1. *90 validate modeled the results --> validate the modeled results*

   **RE:** Thanks, we changed the sentence to '**test the modeled results**' as suggested. (please see line 95)

2. *314 I don't quite understand what the two D values are referring to (D = 0.65, D = 0.56)*

   **RE:** Sorry for my carelessness, we revised as "($D$=0.65, $D$=0.56 for NA-1 and NA-2, respectively)". Please see line 408.

3. *485 pluses --> pulses*

   **RE:** Done as suggested.

4. *485 to captured --> to be captured*

   **RE:** Thank you for this. Revised as suggested (please see line 542).

**Reference**

Chalk, P., and Smith, C.: On inorganic N uptake by vascular plants: Can 15N tracer techniques resolve the NH4+ versus NO3− "preference" conundrum?, Eur. J. Soil Sci., 72, 1762-1779, https://doi.org/10.1111/ejss.13069, 2021.

Cui, F., Yan, G., Zhou, Z., Zheng, X., and Deng, J.: Annual emissions of nitrous oxide and nitric oxide from a wheat-maize cropping system on a silt loam calcareous soil in the North China Plain, Soil Biology & Biochemistry, 48, 10-19, 10.1016/j.soilbio.2012.01.007, 2012.

Daryanto, S., Wang, L., Gilhooly, W. P., III, and Jacinthe, P.-A.: Nitrogen preference across generations under changing ammonium nitrate ratios, Journal of Plant Ecology, 12, 235-244, 10.1093/jpe/rty014, 2018.

Ferrari Machado, P. V., Wagner-Riddle, C., MacTavish, R., Voroney, P. R., and Bruulsema, T. W.: Diurnal Variation and Sampling Frequency Effects on Nitrous Oxide Emissions Following Nitrogen Fertilization and Spring-Thaw Events, Soil Sci. Soc. Am. J., 83, 743-750, https://doi.org/10.2136/sssaj2018.10.0365, 2019.

Foley, J. A., Prentice, I. C., Ramankutty, N., Levis, S., Pollard, D., Sitch, S., and Haxeltine, A.: An integrated biosphere model of land surface processes, terrestrial carbon balance, and vegetation dynamics, Global Biogeochemical Cycles, 10, 603-628, 10.1029/96gb02692, 1996.

Francis Clar, J. T., and Anex, R. P.: Flux intensity and diurnal variability of soil N2O emissions in a highly fertilized cropping system, Soil Sci. Soc. Am. J., 84, 1983-1994, https://doi.org/10.1002/saj2.20132, 2020.

Giltrap, D., Yeluripati, J., Smith, P., Fitton, N., Smith, W., Grant, B., Dorich, C. D., Deng, J., Topp, C. F., Abdalla, M., Liáng, L. L., and Snow, V.: Global Research Alliance N2O chamber methodology guidelines: Summary of modeling approaches, Journal of environmental quality, 49, 1168-1185, https://doi.org/10.1002/jeq2.20119, 2020.

He, W., Dutta, B., Grant, B. B., Chantigny, M. H., Hunt, D., Bittman, S., Tenuta, M., Worth, D., VanderZaag, A., Desjardins, R. L., and Smith, W. N.: Assessing the effects of manure application rate and timing on nitrous oxide emissions from managed grasslands under contrasting climate in Canada, Science of the Total Environment, 716, 10.1016/j.scitotenv.2019.135374, 2020.

Ito, A., Nishina, K., Ishijima, K., Hashimoto, S., and Inatomi, M.: Emissions of nitrous oxide (N2O) from soil surfaces and their historical changes in East Asia: a model-based assessment, Progress in Earth and Planetary Science, 5, 10.1186/s40645-018-0215-4, 2018.

Jones, S. K., Famulari, D., Di Marco, C. F., Nemitz, E., Skiba, U. M., Rees, R. M., and Sutton, M. A.: Nitrous oxide emissions from managed grassland: a comparison of eddy covariance and static chamber measurements, Atmos. Meas. Tech., 4, 2179-2194, 10.5194/amt-4-2179-2011, 2011.

Kronzucker, H. J., Siddiqi, M. Y., and Glass, A. D. M.: Conifer root discrimination against soil nitrate and the ecology of forest succession, Nature, 385, 59-61, 10.1038/385059a0, 1997.

Kucharik, C. J., Foley, J. A., Delire, C., Fisher, V. A., Coe, M. T., Lenters, J. D., Young-Molling, C., Ramankutty, N., Norman, J. M., and Gower, S. T.: Testing the performance of a Dynamic Global Ecosystem Model: Water balance, carbon balance, and vegetation structure, Global Biogeochemical Cycles, 14, 795-825, 10.1029/1999gb001138, 2000.

Lammirato, C., Lebender, U., Tierling, J., and Lammel, J.: Analysis of uncertainty for N2O fluxes measured with the closed-chamber method under field conditions: Calculation method, detection limit, and spatial variability, J. Plant Nutr. Soil Sci., 181, 78-89, 10.1002/jpln.201600499, 2018.

Li, C. S., Aber, J., Stange, F., Butterbach-Bahl, K., and Papen, H.: A process-oriented model of N2O and NO emissions from forest soils: 1. Model development, Journal of Geophysical Research-Atmospheres, 105, 4369-4384, 10.1029/1999jd900949, 2000.

Lognoul, M., Debacq, A., De Ligne, A., Dumont, B., Manise, T., Bodson, B., Heinesch, B., and Aubinet, M.: N2O flux short-term response to temperature and topsoil disturbance in a fertilized crop: An eddy covariance campaign, Agricultural and Forest Meteorology, 271, 193-206, 10.1016/j.agrformet.2019.02.033, 2019.

Malhi, S. S., Nyborg, M., Jahn, H. G., and Penney, D. C.: Yield and nitrogen uptake of rapessed (Brassica campestris L.) with ammonium and nitrate, Plant Soil, 105, 231-239, 10.1007/BF02376787, 1988.

Mosier, A. R., Halvorson, A. D., Reule, C. A., and Liu, X. J.: Net global warming potential and greenhouse gas intensity in irrigated cropping systems in northeastern Colorado, Journal of Environmental Quality, 35, 1584-1598, 10.2134/jeq2005.0232, 2006.

Ogejo, J. A., Senger, R. S., and Zhang, R. H.: Global sensitivity analysis of a process-based model for ammonia emissions from manure storage and treatment structures, Atmos. Environ., 44, 3621-3629, https://doi.org/10.1016/j.atmosenv.2010.06.053, 2010.

Pappas, C., Fatichi, S., Leuzinger, S., Wolf, A., and Burlando, P.: Sensitivity analysis of a process-based ecosystem model: Pinpointing parameterization and structural issues, Journal of Geophysical Research: Biogeosciences, 118, 505-528, https://doi.org/10.1002/jgrg.20035, 2013.

Pfab, H., Palmer, I., Buegger, F., Fiedler, S., Mueller, T., and Ruser, R.: Influence of a nitrification inhibitor and of placed N-fertilization on N2O fluxes from a vegetable cropped loamy soil, Agriculture Ecosystems & Environment, 150, 91-101, 10.1016/j.agee.2012.01.001, 2012.

Reeves, S., and Wang, W.: Optimum sampling time and frequency for measuring N2O emissions from a rain-fed cereal cropping system, The Science of the total environment, 530-531C, 219-226, 10.1016/j.scitotenv.2015.05.117, 2015.

Senapati, N., Chabbi, A., Giostri, A. F., Yeluripati, J. B., and Smith, P.: Modelling nitrous oxide emissions from mown-grass and grain-cropping systems: Testing and sensitivity analysis of DailyDayCent using high frequency measurements, Science of the Total Environment, 572, 955-977, 10.1016/j.scitotenv.2016.07.226, 2016.

Shcherbak, I., Millar, N., and Robertson, G. P.: Global metaanalysis of the nonlinear response of soil nitrous oxide (N2O) emissions to fertilizer nitrogen, Proceedings of the National Academy of Sciences of the United States of America, 111, 9199-9204, 10.1073/pnas.1322434111, 2014.

Thorburn, P. J., Biggs, J. S., Collins, K., and Probert, M. E.: Using the APSIM model to estimate nitrous oxide emissions from diverse Australian sugarcane production systems, Agriculture Ecosystems & Environment, 136, 343-350, 10.1016/j.agee.2009.12.014, 2010.

Tian, H., Xu, X., Liu, M., Ren, W., Zhang, C., Chen, G., and Lu, C.: Spatial and temporal patterns of CH4 and N2O fluxes in terrestrial ecosystems of North America during 1979-2008: application of a global biogeochemistry model, Biogeosciences, 7, 2673-2694, 10.5194/bg-7-2673-2010, 2010.

Tian, H., Yang, J., Lu, C., Xu, R., Canadell, J. G., Jackson, R. B., Arneth, A., Chang, J., Chen, G., Ciais, P., Gerber, S., Ito, A., Huang, Y., Joos, F., Lienert, S., Messina, P., Olin, S., Pan, S., Peng, C., Saikawa, E., Thompson, R. L., Vuichard, N., Winiwarter, W., Zaehle, S., Zhang, B., Zhang, K., and Zhu, Q.: The Global N2O Model Intercomparison Project, Bulletin of the American Meteorological Society, 99, 1231-1251, 10.1175/bams-d-17-0212.1, 2018.

Venterea, R. T., Petersen, S. O., de Klein, C. A. M., Pedersen, A. R., Noble, A. D. L., Rees, R. M., Gamble, J. D., and Parkin, T. B.: Global Research Alliance N2O chamber methodology guidelines: Flux

calculations, Journal of environmental quality, 49, 1141-1155, https://doi.org/10.1002/jeq2.20118, 2020.

Zhang, K., Peng, C., Wang, M., Zhou, X., Li, M., Wang, K., Ding, J., and Zhu, Q.: Process-based TRIPLEX-GHG model for simulating N2O emissions from global forests and grasslands: Model development and evaluation, Journal of Advances in Modeling Earth Systems, 9, 2079-2102, 10.1002/2017ms000934, 2017.

---

## Author Comment (AC2)

Our point-by-point responses are provided below. The *referees' comments are italicized*, our answers are in blue and the **texts from the manuscript is highlighted in bold** for the Editor's easy reference. Some key issue we want to address is underlined:

*This paper describes the integration of agricultural practices into the TRIPLEX-GHG Model and the evaluation and tuning of the model against measurements.*
*It is an enormous project to integrate the great heterogeneity of global agriculture into a global model and to evaluate it. It is clearly a task that needs to be published in steps. The current paper indeed makes some headway in this task.*
*RE*: We thank you for your positive feedback and encouragement! We have tried our best to improve the quality of this study in the revised MS.

*Looking at the paper as a whole, two questions stand out. (i) Does this paper really go far enough at this stage in development to warrant publication? To answer this question the paper should improve its explanation of how it is modified from previous versions. (ii) Secondly, the authors should further explore the underlying physical meaning of the model tuning. Details are given below.*
 *RE*: We found your insights were very helpful to improve the quality of our work and we made every effort to carefully understand and reply your questions and suggestions accordingly.

In order to answer the first question you raised, 1) we simplified the description of the $N_2O$ production processes of the previous version of the model (please see page 5-6 116-132).

2) we improved the description of the new integrated agricultural practices with more specific details and equations (Eq 4-14) as suggested (please see section2.2, page 6-9).

3) In the meantime, we conducted sensitivity experiment of the newly integrated processes to show the effect of these agricultural practices on $N_2O$ flux (please see section3.1.1 line 230-251), which highlighted the difference between cropland and natural soil, current model and pervious version (please see page 13-14, line 366-385).

The sensitivity analysis of model parameters was also revised accordingly to prevent possible misunderstanding (e.g., the subtitle 3.1.2 was changed to '**sensitivity analysis of model parameters**' instead of 'sensitivity analysis') in the revised manusrcipt.

To answer the second question, we conducted the comparison of modelled and reported mean $N_2O$ emission rates and emission factors (EFs, the percentage of total N input emitted as $N_2O$) of the 39 calibrated sites with the continent mean value of the calibrated parameter $COE_{dNO3}$. A reasonable results of EF could support the underlying physical meaning of the model design and calibrated parameters (please see page 17, line 487-500). Furthermore, as you suggested, there are significant difference (ANOVA, $p < 0.01$) among the value of $COE_{dNO3}$ across different continents probably because of the diverse management history and routines of the applied management practices for different continents. These work are designed to show the effectiveness of the improved model and the parameters (please see section 3.2.3 and 4.2.6). The reasonable $R^2$ of the reported and estimated EFs ($R^2 = 0.70$) suggested that improved model with tunned parameter was capable to reflect the response rate of $N_2O$ to external N fertilizer application which is the most important feature of the cropland ecosystems. Hopefully you can agree with our findings.

*1. On my initial reading of the paper it was difficult to discern the new modifications to the TRIPLEX-GHG model from what has already been published. The model description discusses both*

*the older published model and the newer added processes. It would be helpful at the outset to clearly describe exactly what new processes have been added. From my reading they are the following: addition of agricultural fertilizer as either synthetic fertilizer or manure, some global rules for the incorporation of tillage, the addition of flood irrigation, new rules for plant uptake and return of harvested plant residues to soils. As I understand it the nitrification sub-model (and Table S1) is not new but is part of what has already published. If already published as part of a coherent modeling system it is unclear why the paper needs to repeat the model equations here, except to highlight the 13 tuning parameters examined.*

**RE:** Thanks for your suggestion and we were sorry to let you feel in this way. We first re-arranged the Section.2 model description with two subtitles '**2.1 Overview of the TRIPLEX-GHG model v1.0**' (please see line 97) and '**2.2 Model improvement of the effects of agricultural management practices for TRIPLEX-GHG model v2.0**' (please see line 139), respectively to highlight the improvement we made for the model v2.0.  We also added a sentence to further point out what changes we made at the beginning of section 2.2 by saying '**This study improved the model description of plant N uptake and integrated major agricultural management practices including harvest, returned residuals, chemical N fertilizer application, manure application,irrigation and tillage into original model structure as described below in detail**.' (please see page 6, line 139-145). Second, we reduced the context of the description of previous version of the model (please see line 116-132). In addition, we provided more details and equations about the integrated or improved processes (e.g., Eq(6-7)).

Your understanding was 100% correct. Indeed that we included a general description of previous version of the model, especially for the denitrification processes (Zhang et al., 2017). We repeated this published equation of denitrification here because the most sensitive parameter of the model, $COE_{dNO3}$ that we used as the fitting parameter during calibration, is within the denitrification process. We believed that it was reasonable and easier to follow if we presented this parameter in advance than just giving a table or figure with the key parameter that we are going to use jump out from no where. By doing this, we hope to let the audience know what the fitted parameter stands for and where it acts as a key role to the $N_2O$ production. $N_2O$ production processes is critical for our model as a distinct-designed characteristics compared with other global process-based models (e.g., DLEM, VISIT and DAYCENT) (Tian et al., 2010; Ito et al., 2018) and it is of the importance to the model performances (although you pointed out that it was possible that the relatively good modelled result was not necessarily credited to model structure but to the timing of practices and this we will discuss for following questions below). We have therefore simplified and revised by minimizing the text of description of $N_2O$ production (please see section 2.1 Overview of the TRIPLEX-GHG model v1.0). We hope you can understand.

Second, we totally understand your consideration that the new developed processes should be highlighted and can not be mixed with previous version. We provided more detailed description and equations of the integrated processes that were not included in the previous version of manuscript (please see section 2.2 Model improvement of the effects of agricultural management practices for TRIPLEX-GHG model v2.0 from line 139). We apologized for the missing information of the pervious manuscript.

*Some of the newly added processes should be described in greater detail. Flood irrigation is not well described. Is this done continuously or only when soils dry sufficiently, or ...? Is it done*

*everywhere or are rainfed croplands separated from those irrigated. Harvest is also not well described. What are the carbon and nitrogen losses during harvesting? It is not just litterfall that is lost during harvesting but presumably a good amount of the harvested plant is lost to the food system.*

**RE:** We apologize for the missing description of irrigation and harvest. We have added more information associated with the design of the integrated model processes (please see section 2.2).

As for the timing and intensitiy (e.g., Is this done continuously or only when soils dry sufficiently, or …? Is it done everywhere or are rainfed croplands separated from those irrigated), we added more information about the design for model simulation in section 3.2.3 model calibration and validation (please see line 323-337).

**"After spin up, the model simulation was started on January 1$^{st}$, 1901, and ended on December 31$^{st}$, 2015 in daily timestep driven by the daily climate data for each site along with other site-specific input information described in the section 3.2.2. …… In addition, except for harvest which was systematically happening, the timing and management intensities of other agricultural practices were set according to the published literatures and input information...."**

In short, the timing and intensity of agricultural practices are based on the information provided by papers (please see line 333-334). We added this information here because the design matters mostly for model simulation while the section 2.2 is responsible for describing the processes and mechanism, the timing of application of those practices is clearly not a part of 'mechanism' but 'simulation design'. We hope you can understand this. The detailed model processes description of irrigation was shown in line 199-210.

Therefore, to answer your question, irrigation is not done everywhere. It is neither controlled by the input information of the irrigated or rainfed cropland (this dataset is the fraction of particular land use type of each grid cell and is used as site history information). Irrigation was designed to be done in model, when the site experiment information said it was happening. The timing and amount of irrigation is depended on published article for calibration and validation. For instance, NA-8 and NA-9 provided the information of irrigation so that we used these for calibration.

As for harvesting, it happens systematically after each growing seasons. We added detailed description of harvest design in line 158-165 by saying

**"85% and 60% the total biomass carbon (aboveground and belowground, calculated based on the turnover rates in the plant phenology module) was lost via harvest practices for annual and perennial crops, respectively. The loss of nitrogen was therefore calculated based on the C:N ratios of different carbon pools of plant organs".**

Our design is similar with that of VISIT and DLEM, as large scale process-based models, described the harvest by removing all above ground biomass from the system (Tian et al. 2010; Ito et al. 2017). Because the proportion of litter fall to total biomass is large enough for C3 and C4 crops (meaning less biomass is left at the end of growing seasons) so that the effect of removing litter fall is reasonable for simulating harvest. We hope you can understand this.

In summary, we revised this part following your advice (please see section 2.2, page 6-9, line 138-223).

*When building a new model the sensitivity to added processes is usually evaluated. While the paper describes the sensitivity to various parameters it is not clear how the newly added agricultural processes impact the nitrogen cycle. To what extent are nitrification and denitrification processes*

*altered by these processes as claimed in the manuscript? (For example, what is the impact of tilling on the nitrogen cycle?)*

**RE:** Thank you for pointing out this. We apologize that we did not consider the sensitivity of the integrated agricultural practices and it was conceivable that their effects on $N_2O$ emission which we focused on would be evident.

Therefore, we added up the sensitivity experiment of the integrated major agricultural practices on the $N_2O$ flux and the results further confirmed our expectation. The sensitivity experiments were conducted by comparing the model outputs (i.e., annual $N_2O$ emission) between the improved version with one new integrated process for each time and original model to emphasize the 'improvement'(please see line 227-250 and line 364-384). The absolute change ($\Delta N_2O$ emission rate) and relative change (relative effect size) of $N_2O$ emission were measured. Because the targeted variables we were interested in was $N_2O$, we did not evaluate the impacts of these integrated agricultural practices on the nitrification rate exact as you suggested but we believe that by showing the changes in $N_2O$, as an essential indicator of soil N cycle, can indirectly provide insight to the degree of response of nitrification and denitrification. We hope this is acceptable for you.

For your information, the impact of tillage on $N_2O$ emission is described in results section saying

**'the range of the annual $N_2O$ emission response to tillage practices was less evident in terms of the relative effect size (RES) as 0.16. The absolute effect of tillage was larger ranging between -14.82 to 18.27 mg N m$^{-2}$ yr$^{-1}$.'** Please see line 380-384.

*The model description does not adequately describe some of the shortcomings of the implementation from the onset, although the conclusion adds more detail. It seems this new version of the model does not really have a crop model, but the crops are somehow woven into the existing pfts. For example, the paper states over Australia that PFTs with tropical forest and shrub instead of agricultural lands were used.*

*Moreover, while the emissions of N2O from specific types of crops are measured, only C3, C4 and rice are used as crop types in the model. How does this comparison work? It seems probable that crops do not have their own soil column although this is not clear. The paper needs to state these considerations at the beginning.*

*Far more detail is needed into how crops are integrated into the pft structure of the model.*

**RE:** Thanks for your pointing out of these possible questions associated with model structures and simulation processes. We agree with you!

First, it is very true that we added the discussion of shortcoming of the current model design in the discussion section in the revised MS (please see lines from 593 to 633). That was because these problems were for the improved model and were found after all of the model sensitivity experiment, calibration and validation. The most evident shortcoming of the previous model was that it can not provide estimations of cropland $N_2O$ (please see line 87-90, 134-138) which was the main problem that we wanted to overcome with this current study. Therefore, we put the shortcomings of the improved model as the source of uncertainties in the discussion section to make sure the logic flow reasonably. Hopefully you can understand our consideration.

Next, we totally agreed that the description of the model simulation was limited. And we improved this part by adding more details as requested including the vegetation design and

agricultural management design of simulation (please see line 324-339). Particularly we emphasized the model design of crop types:

> 'Cropland as one of the 16 vegetation types from input data was further categorized into the plant functional types (PFT) of generic C3, C4 crops and rice based on the local climate as a common practice for large-scale process-based models (Monfreda et al., 2008; Tian et al., 2019). For the experiment sites cultivated cash crops (e.g., sugarcane, litchi and grapes) which have diverse phenology and physiology characters than cereal crops, the PFTs were set manually during the site-based simulation.'

For more information, the new version of the model have 14 plant functional types and 16 vegetation types (1: tropical evergreen forest / woodland; 2: tropical deciduous forest / woodland; 3: temperate evergreen broadleaf forest / woodland; 4: temperate evergreen conifer forest / woodland; 5: temperate deciduous forest / woodland; 6: boreal evergreen forest / woodland; 7: boreal deciduous forest / woodland; 8: mixed forest / woodland; 9: savanna; 10: grassland / steppe; 11: dense shrubland; 12: open shrubland; 13: tundra; 14: desert;15: cropland; 16: polar desert / rock / ice). Vegetation types are imported from input data. As for the cropland, it can be further divided into 2 PFTs, C3 crop and C4 crop, based on climate calculated within the model. You mentioned that the forest and shrubland PFT instead of all were used for a few site-level calibration. That was due to the fact that the plant physiological characteristics of sugarcane and leechee etc. are way different with that of C3 and C4 crop (annual crop mostly). For instance, the size of wood biomass pool of C3 and C4 crop are limited (as defined in the plant physiology submodule) while wood accounts for a large proportion of total biomass of shrub and tree (perennial crops). The difference results in large divergence of biomass allocation strategy, N requirement, biomass turn over rate and the properties of returned straw (Zhang et al., 2017; Kucharik et al., 2000; Foley et al., 1996).

You further brought out a question that, how the simplified category of crop types (C3, C4 crop and rice-paddy) can support the modelled reliability of $N_2O$ from different croplands? It is a critical question to current modelling of large scale $N_2O$ emissions. Admittedly this can lead to the uncertainties of the model while the statement that 'Vegetation types have varying impact on soil $N_2O$ emissions' was not solid according to current field work. In short, crop or vegetation types might have minor (more or less neglectable) impact on the $N_2O$ emissions without the existence of legume species while this question should be paid more attention by large-scale process-based modelling study in the future. We included this issue in discussion section (please see page22, line 620-625).

To our knowledge, most studies focused on the anthropogenic management effects on $N_2O$ flux rather than crop types. It is a clear sign that at least management plays a more important role in controlling $N_2O$ flux than plants (Shcherbak et al., 2014). Vegetation or crop types affect the N cycling mainly by different N uptake abilities. For instance, in comparison with annual crops, perennial crops likely affect the $N_2O$ emissions by increasing competition for available $NO_3$ (Rochette et al., 2018). However, on one hand, since excessive external input N to cropland soil, there is no N-limitation to crops growth and the surplus N would be therefore emitted as N-trace gases (Shcherbak et al., 2014). On the other hand, there is no significantly large difference of the N uptake abilities within C3 (wheat, cotton) and C4 (maize) broad categories.

Previous studies proposed that the factors that significantly influence agricultural $N_2O$ emissions were N application rate, fertilizer type, soil organic C, crop type, soil pH and texture(Stehfest and Bouwman, 2006; Bouwman et al., 2002; Rochette et al., 2018). Using the same

data source, random forest (a machine learning technology) obtained a similar conclusion that crop type can affect the $N_2O$ emission pattern from fertilized soil (Philibert et al., 2013). For example, according to Rochette and Janzen (2005), legume crops generally result in lower emission levels than other crop types (probably because of lower N fertilizer application rate for legume with strong biological N fixation). However, we must realize that several management practices are adjusted as a function of the type of crop grown (e.g., soil tillage) and because some crops are cultivated only on particular soil types or in particular climatic conditions (e.g., wetland rice mostly grown in poorly drained soil) (Philibert et al., 2013). If we exclude those 'special crop types' (e.g., legume species), that would be a totally different story. A global meta-analysis strongly supported this assumption, only N-fixing crops had a emission factor (EF) larger than the other crop groups (upland grains, rice, and perennial grasses) while there was no statistically significant difference among other crop types (even between Forage / Perennial grasses and annual upland crops) at global scale (Shcherbak et al., 2014). Recent study found a result that no significant impact of vegetation species on $N_2O$ emission (Baskerville et al., 2021). In addition, another meta-analysis demonstrated that no significant effect of cover crop on $N_2O$ emission was detected (Mohamed et al., 2019). These founding again supported that except legume species, the crop have minor effect on $N_2O$ emissions which makes the broad category of the crop types of our model more reasonable. Of course, the strong impact of legume species is important for modelling and should be further improved. We discussed about this briefly in discussion section to show that the current model design of vegetation is a potential source of uncertainty for modelling (please see line 618-625 and line 655-657).

There was also technological problem associated with the modeling of the historical $N_2O$ emission at global scale with specific crop types. In the historical periods, the geographic distributions of different crop types are rarely investigated (recent remote sensing studies have interests in identifying the plant species) because instead of climate regions, the cultivated crop species are controlled by individual farmers with strong variability or randomness (Monfreda et al., 2008). This is another reason that a majority of the large scale process-based model used C3 and C4 crop types instead of specific crop types which have been proved reasonable (Tian et al., 2019; Ito et al., 2018).

*Is the added nitrogen from fertilizer apportioned to the crops or is it apportioned to the whole grid square? How do the nitrogen demands of the crops interface with the other vegetation within each model grid?*

**RE:** Thanks for your good question. For sure, the applied N fertilizer is apportioned to the cropped area only. Before 1961, all vegetation type were used based on the input vegetation map for simulating the soil C and N dynamics (on site level, or in other words, in one grid cell) because of limited anthropogenic external N input before 1961. During this time, the nutrient and water demands were comparable for different PFTs in a same grid cell. As we described above, more than one PFTs can be found in a same grid cell which is calculated within the model from input vegetation map. Cropland fraction and grassland, forest land and bare land fraction were considered in the model structure to provide a weighted mean of the different soil nutrient pools. After 1961, with increasing external N input to the cropland soil, the model considered the whole grid cell as cropland to avoid possible under- or over-estimation of the soil inorganic N level. And all the added N fertilization was apportioned to the whole grid cell. Therefore, the modeled results (e.g., $N_2O$ emission, C and N dynamics) did not represent the whole grid cell, but represented the condition of

croplands in the grid cell. That was the reason that we incorporated the cropland fraction of each grid cell into our model.

However, we should notice that it is possible that there are transformation of land use or land cover (from cropland to conservation land or vice versa) while fraction of cropland may not change at all. In our consideration, during the historical period of time, there is a significant growing trend for cropland area from 5% in 1860s to 12% in 2000s of the total terrestrial area. Limited conservation or restoration activity happened until recent decade like the 'Assessment of Conversion Cropland to Forest and Grassland Project' in China. Therefore, the uncertainties of the model estimation related with the statement above is acceptable (although it was not quantified).

The question you raised above, is very important and it may bring some uncertainties to our estimation as a part of experiment history. We summarized and discussed the possible uncertainties in the discussion section (please see line 626-634).

**"In addition, the uncertainties in the site history are also responsible for the model inaccuracy because the historical management has a tremendous effect on the soil properties and C, N dynamics (Gelfand et al., 2016; LaHue et al., 2016) with strong legacy effect on $N_2O$ emissions (Liu et al., 2014; Jiang et al., 2018; Zhou et al., 2017). For instance, …… because these agricultural practices are controlled by the individual farmers and vary greatly at the local and subregional scales, without clear global distribution patterns such as those for soil and climate (Wang et al., 2018b). The insufficient reported site management history therefore set a barrier to the accurate estimation of the local soil nutrient conditions and thus $N_2O$ emissions."**

*Related Text:*
*l223-224: How were data transformed to spatial resolution of the model? Does this mean each 0.5x0.5 grid only includes one crop or are other pfts mixed in? How does this fit into the vegetation dataset? Is there a separate agriculture model? How were specific crop measurements evaluated within the model structure?*

 **RE:**   Thanks for your good questions.

First, the data were transformed by the ArcMap software with the tools 'aggregate'. We thought this was a common practice for modelling and ArcGIS so that we chose to simplify the description.

Next, not exactly, more than one PFTs can be found in one grid cell. For example, in a grid cell in eastern America, deciduous broadleaf trees, temperate conifer evergreen trees and cold shrubs and cool grass simultaneously (Peng et al., 2013; Kucharik et al., 2000). The PFTs are calculated within the model (vegetation dynamics module) based on the input vegetation dataset and land use fraction (see answers above and the line 328-333).

**"No specific crop types (e.g., maize, wheat and soybean) was included for current model. Cropland as one of the 16 vegetation types from input data was further categorized into the plant functional types (PFT) of generic C3, C4 crops and rice based on the local climate as a common practice for large-scale process-based models (Monfreda et al., 2008; Tian et al., 2019)."**

The input dataset is upstream, giving the information of vegetation types (16 of them) to the model and model generates the PFTs of each grid cell according to the input vegetation types and climate variables (Zhang et al., 2017; Kucharik et al., 2000). However, after 1961, the current TRIPLEX-GHG model considered all grid cell be covered by cropland (C3 or C4 crops) to avoid

possible large uncertainties associated with soil C, N dynamics. Therefore, the model output represents the condition of cropland within this grid cell. Hope you can understand our design here.

There is no separate agricultural model available for this version of TRIPLEX-GHG. C3, C4 crops and rice are PFTs calculated from the vegetation type 'cropland'.

If I understand your final question correctly, the answer is as follows. The crop types included in current model structures are C3, C4 crop and rice-paddy. As shown in the Table 2, wheat, as a mostly used C3 crop is widely cultivated in China and Europe. Corn, the commonly cultivated C4 crop is grown mostly in great lakes region and Africa. Results (calibration results) suggested that the model design was reasonable.

*l163. What is the cropland ecosystem? Have crops been added specially?*

 **RE:** Thanks for your question.   Cropland ecosystem means the cropland area of the grid cell. We changed the term to 'cropland area' as suggested (please see line 160). Crops were not specially added.

> **"We systematically removed all of the litterfall from the cropland area of the grid cell at the end of the growing seasons to modify the harvest."**

Again, we should highlight that, the cropland area means the total grid cell area after 1961 because from 1961, the simulation is designed based on the assumption that this grid cell is covered by only cropland.

*2. Equations (2) and (3). Is the plant N demand for all plants changed or just for crops alone? The extent to which crops and plants prefer taking up nitrate is quite a strong assumption, not universally true, and contrary to many models. The references included to justify this assumption are rather old. This warrants more discussion as the results are likely to be very dependent on this assumption.*

 **RE:** We appreciated for your questions associated with model structures. Yes, this plant N demand is applicable for all soils not just crops alone (please see supplementary material Table. S2 showing that the improved model description of plant N uptake improved the model ability in modelling $N_2O$ emission from natural grassland soil). For crops? Sure! And for managed shrublands and cash crops (e.g., banana and leechee like for validation and calibration)? Also true!

The understanding of the plant preference of the form of N uptake has been changing with advance technologies. Such preference have been studied and found to have large divergence across different plant species, growth stage and soil texture or properties and can shape the function of local ecosystem (Boudsocq et al., 2012). At first, ammonium ($NH_4^+$) was thought to be preferred by plant because of the lower energy requirement to absorb them which can be directly incorporated into glutamate via an $NH_4^+$ assimilation pathway. However, some studies also revealed that only $NH_4^+$ can cause severe toxic symptom. The $NH_4^+$ toxicity could counterbalance the energetic advantage of $NH_4^+$. After 1990s, more evidence demonstrated that the uptake rate of $NO_3^-$ is higher than expected and might dominate the total N uptake in some circumstances. And the recent tracer results suggested that plants prefer nitrate to ammonium even though the conclusion may be challenged by interpretation of tracers. A safer and more objective statement would be: plants exhibit plasticity rather than preference in the acquisition of ammonium and nitrate (Chalk and Smith, 2021)

In general, $NO_3^-$ is usually more available for plants, owing to its higher mobility which leads to more rapid diffusion to root and easier access to plant as mass flow and diffusion is main pathway for N uptake (Daryanto et al., 2018). In well aerated agricultural soils or other frequently disturbed

sites, $NO_3^-$ is the principal inorganic N source. According to Tisdale et al. (1993, Soil Fertility and Fertilizers), the rate of $NO_3^-$ uptake is usually high and is favored by low-pH conditions. $NH_4^+$ uptake proceeds best at neutral pH values and is depressed by increasing acidity. Marchner (1995, Mineral Nutrition of Higher Plants), states that when both forms of N are supplied, it is easier for the plant to regulate intracellylar pH and to store some of the N at low energy costs. In the continuously cultivated wheat cropland fertilized by ammonium-nitrate, a subsidiary experiment found more N was retained in the soil at harvest when the fertilizer was added in the ammonium form than as nitrate; 32.6 and 19.5 kg ha-1, respectively (Shen et al., 1989). Another grassland study observed a comparable recovery rate for $NH_4$ and $NO_3$ (Jenkinson et al., 2004) and in alpine region, managed grasses were found to prefer $NH_4^+$ at the early stages but switched the preference for $NO_3^-$ later (Cui et al., 2017). Long-term field study also revealed that more than 60% of the applied nitrate was uptake by crops with isotopically labeled nitrogen fertilizers (Sebilo et al., 2013). All experiment suggested that at least a comparable preference of the $NH_4$ and $NO_3$ is required in fertilized soil.

As you pointed out, the design are likely to have significant impact on the model output (i.e., $N_2O$). Well, according to the sensitivity experiment, the difference was not as obvious as that of managements. This feature further highlighted the significance of the management effect on cropland $N_2O$.

Li et al. (2000) first proposed this way that setting a higher priority of $NO_3$ uptake to get a reasonable answer of soil    $N_2O$ production (Li et al., 2000). The original model design for plant N uptake was that root uptake $NH_4^+$ first to satisfy the plant N demands (Foley et al., 1996; Kucharik et al., 2000; Zhu et al., 2014; Zhang et al., 2017), this design is also adapted by other process-based models (e.g., VISIT Ito et al. 2017, DLEM Tian et al. 2010). However, given the mechanism of plant N uptake process, this phenomena results in that no-nitrate can be uptake by plant which is not practical and not realistic. $NO_3^-$ is supposed to have a higher priorate for plant, at least than that of previous version of the model. Plus, considering the higher external nitrate source to cropland soil than that of natural soil, it is critical for the model to prevent an un-excepted high soil $NO_3$ level which is likely to cause overestimation of denitrification, thus, $N_2O$ emission.

We hope you can understand our design and the result were found reasonable. We added more reference to support this assumption and rephased to

**'In cropland soil, $NO_3^-$-N is more easily absorbed by roots due to higher concentration and mobility'** (please see page 6, line 148-149).

*Despite the rather simplified representation of agricultural practices the tuned model reproduces the measurements with remarkable fidelity. While the authors list many aspects of the agriculture that could be improved, it is difficult to see how they could improve on their present results with the metrics used (R2=87%!) It seems like there might be two possibilities: (i) the precise representation of agriculture is not important or (ii) there is something about the tuning procedure that allows the model to get the right answer. My guess it is the second possibility. If that is true, the model won't show much sensitivity to the parameterization of agriculture or the parameterization is irrelevant as the tuning parameterization will easily compensate (see (1) above). This would be easy to check. One hypothesis is that to a large extent the model solution is dictated by the timing of the fertilizer input. As the plants do not take up ammonia preferentially, most ammonia added is quickly nitrified*

*to nitrate. With this assumption nitrate acts as the crucial pivot point controlling N2O production making it rather easy to tune. Real model skill could best be assessed by looking at other aspects of the simulation: for example, the emissions per fertilizer added (the emission factor), the interannual variability in the emission factor at a particular station or the difference in emission factors between stations. The D value and RMSE with which the model is evaluated would seem to emphasize getting the maximum values correct which seems highly dependent on the amount of fertilizer added. More interesting would be to assess skill in predicting the emission factor or other aspects of the simulation.*

**RE:** We thank you for your questions and your concern associated with the model improvement, model structure and the way to evaluate the model performances. It is a big task and challenge to answer your questions and explain the standing points of ourselves. We response point-by-point here.

First, it is not contradicted for a relatively good agreement between modelled and observed daily mean flux during the field experiment periods and model improvement in the future. As you suggested, we provided a comparison between modeled emission factors (EF) and observed EF of 35 calibrated sites (the remaining 4 were not included as they did not report EF or design a control plot). We found a less constrained regression results of EF ($R^2$ = 0.70, slop = 0.72) compared with that of emission rates ($R^2$ = 0.87, slop = 1.07) (please see line 486-499). And we discussed this phenomena and hypothesized that there were further work to be done to improve the underlying mechanism of the model to have a better performance in EF comparison. In the meantime, the discrepancy between modelled and measured variations of daily $N_2O$ flux also required further improvement. We reorganized the discussion of the source of model uncertainties in the new version of the manuscript (please see page 592-648).

Second, you proposed two possible ways to explain the current fide validated results. We don't want to disagree with you. First, compared with previous large-scale process-based model assessments, the current TRIPLEX-GHG model v2.0 showed probably more detailed description of the management practices (to our knowledge, although not as detailed enough) in varying time steps. Next, the tunning parameter was extremely important for the model to get a right answer in terms of the daily variation of the $N_2O$ flux. The parameter $COE_{dNO3}$ showed strong sensitivity controlling the $N_2O$ emission rates as suggested by sensitivity analysis of model parameters (Fig. 3). The calibration experience suggested that $COE_{dNO3}$ not only controlled the emission rates (e.g., peak values) but also can change the total emission pattern (Probably because the soil C and N dynamics are changed. Higher $NO_3^-$ consumption rate as controlled by $COE_{dNO3}$ can consume soil DOC to a lower level before experiment year which further alter the emission patterns and emission rates. Such results again emphasize the importance of site history to the model accuracy). We provide additional 3 figures comparing the calibrated parameters and continent mean parameters (i.e., continent mean value of EU, AS, NA = 0.0294, 0.021, and 0.0299, respectively) effects on the daily $N_2O$ variation below.

[Figure]

[Figure]

[Figure]

The results suggested that: 1) the parameter $COE_{dNO3}$ is extremely important and sensitive to the model output, $N_2O$ flux; 2) the reasonable validated results (measured and modeled daily mean flux during the field observation) do not necessarily mean that the temporal variation of $N_2O$ fluxes are as good as calibrated results for every sites. The outcome again emphasized the significance of further improvement of model.

However, we totally agreed that the timing of fertilization is of the importance for our modelling. But it is essential to point out that, multiple factors of the modeling, including input information (e.g., timing of fertilizer application) and description of underlying mechanism of the model (e.g., features that we proposed in the discussion) controlled the accuracy of the modelled

flux in daily step, thus, cumulative emission. The importance of those two is hard to separate.

Yes, it is true that timing of fertilizer application is critical for modeled accuracy and fertilizer applications are likely to induce $N_2O$ emission pulses quickly. Hence, the increasing amount of N input result in higher $N_2O$ emission peaks and total emissions. But, it is not the universal fact. As a reliable model, TRIPLEX-GHG well reflect the different degrees of the response of $N_2O$ to applied N. Most of the fertilizer application induced the $N_2O$ pulses immediately (or for several days) as suggested by our model results. While there were some exceptions. For instance, the fertilizer was applied in July for the observation conducted in the wheat field Australia (Fig.6 e AU-5) while the emission peaks occurred in few months later until a significant amount (425mm) of rainfall received (in Oct. or Nov.) (Wang et al., 2011). It was fortunate that the resolution of our input meteorology data was fine enough but the most important thing is that thanks to the anerobic concept, the TRIPLEX-GHG model can provide a reasonable estimation of the occurrence of the marked emission peaks, which supported the first feature that the anaerobic balloon concept contribute to the model performance we proposed (Butterbach-Bahl et al., 2013; Song et al., 2019).

Emission factor is the response rate of the soil to external N input, which is determined by soil properties (texture, pH, SOC etc.), physical conditions (temperature and moisture) and nutrient conditions (Bouwman et al., 2002). That's why several emission factor based models integrated other factors such as WFPS, temperature and bulk density (Wang et al., 2019; Zhou et al., 2015). Moreover, EF is not constant for one site but varying with management intensities (the amount of applied N). The sensitivity of $N_2O$ response to N input grows exponentially when the soil N availability becomes larger than plant N demand (Hoben et al., 2011; Shcherbak et al., 2014). The comparison between modeled and measured emission factors (EFs) was considered initially in our study. However, not all of the field studies reported the EFs as it requires continuous observations (some of the studies only conducted the measurements periodically) and the control plots (some of the studies compared the different effects of the different fertilizer without control groups).

Now, we conducted the comparison between modelled and measured EFs thanks to your suggestions since this comparison results is the great way to show the fact that the model accuracy is not only controlled by input management information, like the amount of N fertilizer application, but also largely determined by the well-described underlying mechanism of $N_2O$ production under varying environmental and management conditions (although Zhang et al. (2017) has proved it for natural ecosystems). It is a great method to show the reliability of the model and the results we added confirmed such statement (please see line 486-499). However, we did that for those calibrated sites only and not for validation sites because we have to admit that by comparing the modelled and measure $N_2O$ emission rate instead of the degree of response to N addition is a more direct method to show the model performance since the continent mean parameters were proved applicable for calibration. We hope you can understand this.

*3. The authors use their present results to justify the physics in their model. Thus the authors state that (see L 420-441) the fidelity of their model is "…. derived from three features of our model" (line 423). Because the simulation gets the $N_2O$ peak following fertilization with tuned parameters does not imply these three features are important. To make this statement the authors need to do considerably more work and to use other metrics to justify their model. I don't find these conclusions justified based on the current paper.*

**RE:** We thank you for your constructive suggestion. We agree that these statements in the previous

version of the manuscript was lacking of scientific support. Therefore, sensitivity experiment of the incorporated processes was conducted to show the response rates of the model to those processes to show the effectiveness of the new model design (please see page 14, line 365-384). Afterwards, we compared the modelled and observed Emission Factors (EFs) of the calibrated model to further support the model performance and reliability (please see page 486-499). In the meantime, we added exact examples from calibration results to reflect the model features that we proposed in the discussion section which we have reorganized (e.g., please see line 543-545 )

> **'The TRIPLEX-GHG model v2.0 was capable to reproduce the immediate (e.g., Fig. 5c and Fig. 5e) and postponed (e.g., Fig. 4f and Fig. 7d) responses of fertilizer applications because both the soil oxygen conditions and the soil water conditions were considered'.**

Please kindly check the added results about sensitivity experiment and calibration (section 4.1 and section 4.2.6).

We also reorganized the discussion section by first examining the effectiveness of the integrated processes and the property of selected, most sensitivity parameter (please see page 18, line 517-537) and then by pointing out the advantages of the model design and development as suggested by the calibration results (please see 538-580). The comparison of modeled EF and observed EF were further discussed (please see line 582-591) and we also discussed potential sources of uncertainties and ways to improve the model (please see line 592-660).

*4. The paper title seems to imply the authors are simulating global cropland $N_2O$ emissions. Please give global emission estimates from the model and evaluate the distribution against other published estimates. Alternatively the authors could rephrase to emphasize the fact that only select locations have been evaluated (albeit on most continents) and tuned against.*

**RE:** Thanks! We did have some global emission results but we are expecting to report that in ongoing separate article. As suggested, we changed the title of this paper to

> **'Integrating Agricultural Practices into the TRIPLEX-GHG Model v2.0 for Simulating Global Cropland Nitrous Oxide Emissions: Model Development and Evaluation with Site-level Observations'.**

*5. Also, if I understand correctly the tuning parameters used are different on every continent. It is admittedly rather strange for a process-based model to use a different tuning parameter on every continent. This is not based on any environmental variable or other physical aspect. This feels rather unsatisfactory to use in a global model and needs further justification. Otherwise it seems like the introduction of an arbitrary tuning parameter.*

**RE:** We thank you for your question and we understand your concern. In fact, it was not ideally suited for the global model to have different parameter values for different regions, although several previous studies did similar procedure like Zhang et al. (2017) who used the mean value of the parameters of the same climate regions in a global model.

For this study, large variations were found for the calibrated parameter across the globe. As you mentioned, we did test the possible relationship between parameter values and local environmental input information (e.g., SOC content, total N, clay, sand content, soil C:N ratios etc.) by using simple linear and non-linear regression, random forest and support vector machines etc…, but found no significant results (for sure that this may be due to the limited number of sites, n=39). We therefore hypothesized that this parameter, coefficient of nitrate consumption rate, may not be

controlled by environment instead of anthropogenic factors. Fertilizer application to soil has been proved not only to supply the substrate for the $N_2O$ production processes but also alter the processes (Cui et al., 2016; Yang et al., 2017). Globally, the changes in soil denitrification increased exponentially when the rates of synthetic N fertilizer application ≤ 250 kg N ha−1 (Wang et al., 2018a) which was in line with the exponentially changing $N_2O$ emission rates and EFs (Shcherbak et al., 2014; Hoben et al., 2011; McSwiney and Robertson, 2005). Besides fertilization rates, fertilizer types may also have impact on the denitrification and nitrification processes by affecting community composition and activity of soil microbes (Hu et al., 2015; Yang et al., 2017; Cui et al., 2016; Li et al., 2020). We further test the relationship between the parameter values and applied amount of N, $NH_4$/$NO_3$ ratio but found no significant results. This could be explained by uncertainties generated from other site management history like tillage and vegetation transformation (Wang et al., 2021). Such feature is reasonable considering the strong variation of the agricultural managements are controlled by the individual farms without a clear geographical distribution such as climate variables.

Therefore, we assumed that in a same region or continent where similar routine and habits were applied to the agricultural managements might be a potential way to upscale the calibrated parameters. We used one-way ANOVA to examine the calibrated parameter results among different continents and found significant difference (Table S5). Therefore, the continent-mean parameters were used to further test the model and the results were acceptable as suggested by Fig. 9a-b and Fig. 10.

We think that this parameter is not an arbitrary parameter but actually means something (at least in line with our assumption that this parameter might be controlled by historical managements) which we do not know yet since the reaction rate of the first step of denitrification is hard to be measured separately and directly(Yu and Elliott, 2018). As a theoretical rate, the parameter could be further tested with the field or laboratory studies to examine the activities of enzymes, expression of functional genes. We hope that you can understand this situation here and we also open for other possible methods to up-scale the parameter reasonably at global scale.

*6. It would be helpful to understand how the 13 sensitivity parameters control denitrification. Some of these parameters don't seem to be included in S1. This is where a clear link between the model equations and these parameters would be helpful.*

**RE:** Thanks for your suggestion. We revised the text of the Table S1. The more detailed description of the meaning and the effects of the parameters listed in sensitivity analysis of parameters can be found in Table.1.

As you pointed out that some parameters did not include in Table S1, I believe you were mentioning ($EFF_{NOX}$). These parameters were in the Eq.(1) the most important equation for denitrification processes. We double checked that all the parameters were listed and described.

*7. Figures 3-7. It is not clear what is being shown here (unless I missed it). Please state clearly in the figure caption and in the text. Are these showing measurements against the fitted model solution at each fitted site? Are these showing measurements and the model solution with the continental average of the fitted parameter for the fitted sites? Are these showing the unfitted sites with the continental fitted parameter. It is really the latter which should be shown – otherwise the authors are just showing the tuned results.*

**RE:** Sorry for the missing information. Yes, the Figures showed the measurements against the fitted model solution of the 39 calibrated sites (the tunned results as you said) as we stated that

**'For model calibration, we adjusted the most sensitive parameter of the N₂O emissions in order to fit the best model performance by comparing the output of daily N₂O flux data with the observed data obtained from published papers'** (please see line 339-341).

We revised the figure caption

**'Comparison of the calibrated model results of N₂O emissions against field observations from the cropland sites located in … for model calibration'**.

Moreover, following your further suggestions, we added the site-calibrated values of the tunned parameter, COE$_{dNO3}$ in the figures which also helped to avoid possible misunderstandings. As we stated in the method section (line 339-341), we calibrated the model and parameter to get the best performance of the model. The continental average of the fitted parameters from calibration results were used for comparing the averaged daily emission rates and emission factors during the experiments of the calibrated sites to confirm the reliability of the calibrated model (but were not used to show the daily variation as Fig. 4-8). The continent mean parameters were therefore used for model validation which compared the observed daily mean N₂O flux from 68 field sites with modelled estimations. These 68 sites were not fitted and were excluded from calibration process. Comparing the cumulative emission rate is the same to compare the mean daily emission rate during the experiment period and the latter is even more appropriate with constant unit (mg N₂O-N m⁻² day⁻¹) because a large number of the measurements were not done continuously (e.g., growing seasons only for multiple years).

*Some minor comments:*
*l88 "agricultural practice modules" – it is not clear what is meant here.*
**RE:** Sorry for the phrase. We changed the first objective to

**'(1) to integrate major agricultural practices into the framework of an extant process-based model'** (please see line 92-93).

*Table S1. If retained, please make sure that all the terms in S1 are given. I found a number of symbols and parameters that were not specified.*
**RE:** Sorry for the missing information. As suggested, we revised this part accordingly (please find supplementary material Table S1).

*It is unclear from the model description how soil nitrogen loss to N2 is handled. Could the authors clarify. 150, N2 should be a major gaseous N loss from agriculture.*
*Are atmospheric resistances used to parameterize the flux to the atmosphere?*
**RE:** We apologize for the missing N2. To simplify the description of the model, we deleted the sentences (please see line 147).

The detailed descriptions of the processes of N₂O emission, nitrification and denitrification, were presented in a previous paper of TRIPLEX-GHGv1.0 (Zhang et al., 2017). Since we have revised and shortened the model description section (2.1) as suggested (to highlight the updated part of the model), we added more information in supplementary material.

In short, the denitrification processes in a stepwise reduction process. The final step is to reduce the NO to N₂ with DOC and related denitrifier (Eq. 1). Limited process-based model considered the

$N_2$ emission as the major target variable although they used a ratio '$N_2O/N_2$' or '$N_2O/(N_2O+N_2)$' as a parameter to constrain the $N_2O$ flux (Thorburn et al., 2010; Tian et al., 2010; Ito et al., 2018; Parton et al., 1996). and that direct N2 emissions are difficult to measure, model and hence scale up (Groffman et al., 2009). Few studies focused on the N2 emission and modelling from cropland (Wang et al., 2020). Given the relatively well-constructed description of denitrification, the TRIPLEX-GHG has a great potential to model the flux of N2 with proper calibration and parameterization.

As for the atmospheric resistances, if I understand correctly, it was not included in the calculation of $N_2O$ flux. But, the model decide the $N_2O$ flux with the Fick's law of diffusion which considered the $N_2O$ concentrations of different soil layers and the height, properties of soil profile.

*Please make sure all abbreviations are spelled out: e.g. NOx and DOC are not given.*
 **RE:** Sorry for the carelessness. We added the '**dissolvable organic carbon**' for DOC and '**nitrogen oxides**' for NOx in line 123-124 and 125, respectively.

*l161. It is stated that the supplemental figure referred to proves the effectiveness of the design. First, it is very difficult to evaluate this figure with the information given. What does this show? What are the blue dots and red lines? What are the conditions simulated? Was fertilizer added? It is difficult to see how this figure alone really proves anything.*
 **RE:** Sorry for the limited description and missing legend for this figure. We replaced this with another Figure to show the effectiveness of the modeling of soil concentration of $NO_3^-$ with the improved plant uptake modules.

*How is biofixation handled?*
 **RE:** Sorry for missing biofixation information. The biofixation of N is within the scheme of the original model TRIPLEX-GHGv1.0 (Zhang et al., 2017) calculated as a function of total biomass and PFTs (Tian et al., 2018). Biological N fixation provide extra N for soil, especially for the natural, undistributed ecosystems like amazon rain forests (Tian et al., 2019). However, the contribution of biological N fixation to agricultural soil $N_2O$ emission is minor compared with the excessive and easier accessible inorganic N source from fertilizer application. Except the existence of legume species. However, as we stated before, we do not have specific expression and classification of plant so that the strong bio-fixation of N of the legume species was not integrated in the current model and we will definitely improve this in the near future. Hope you can understand this.

*What is assumed for fertilization timing, harvest timing, tillage timing, irrigation timing?*
 **RE:** We apologize for the missing information. We improved the description of the model setup for simulation processes under the method section 3.2.3 line 324-339.

During the site-level model calibration and model validation, all the management information is based on the published literatures so as the timing of those managements. All of the published articles provide the exact timing and amount of fertilization and irrigation. So that we can just follow those. To be mentioned, only a few sites provided exact time (date) of tillage activity and for those did not, we chose the measure results from non-tillage sites to compare with the model output. The harvest timing as we stated before, is happened systematically at the end of the growing seasons.

*It is not clear what governs the crop demand for nitrate versus the demand from denitrifyers? How are these partitioned?*

**RE:** Thanks for your good question! It is interesting and important. In our model design, the source code showed that the plant N uptake is superior to other soil biogeochemical processes (e.g., denitrification). This design is quite reasonable.

(McSwiney and Robertson, 2005) for the first time found that the distribution of $N_2O$ fluxes across N addition gradient in cropped soil appears to be driven by the patterns of soil N availability. The plant N demand was firstly satisfied (as suggested by reaching the maximum grain yield) before the remaining N could be exported as $N_2O$ (utilized by denitrifiers), which is the mechanism of the non-linear exponential pattern of the $N_2O$ emission from surplus N. Similar results were widely obtained for various cropland ecosystems (Hoben et al., 2011; Shcherbak et al., 2014). Due to excessive external N was applied to cropland soil, the crop growth is not limited by N availability which is the purpose of fertilization. Plus, plant also takes a large amount of $NH_4^+$. Therefore, the competition between denitrifier and plants for soil mineral N (nitrate) is minor and neglectable for cropland model.

To sum up, for the cropland soil where, the soil N is way beyond the combined requirement of vegetation and soil biomes after plant uptake as we designed the model.

We added sentences describing the model design to highlight this critical point (please see line 156-158).

*l267: how were sites chosen for model calibration? At random?*

**RE:** Good questions! We collected all available sites with cropland $N_2O$ emissions (107 sites) for both model calibration (39 sites) and validation (68 sites). We randomly chose 30% of these (32 sites) for calibration (originally the sites for calibration and validation is 3:7 which is a common practice for testing models like random forest and ANN). However, due to the limited high-quality and relative long-term (because calibration need to compare the modelled and observed data in daily time step) reported $N_2O$ data in some developing regions (e.g., Africa and South America), the number of selected sites were limited for these regions to obtain a reasonable calibrated result. Therefore, there are 7 more sites which have higher frequency of measurement were further included into model calibration (39 sites in total). We hope you can understand this procedure.

*Are the fitting parameters significantly different between the continents? Can you give them in the supplement?*

**RE:** Thanks for your question. ANOVA test showed that there were significant difference of the $COE_{dNO3}$ among different continent (F-value=3.971, p-value=0.00627 <0.01). But, the multiple comparison with TURKEY-HSD found that this significance was attribute to the South America, Africa, Australia and Africa although the repeat (samples) for each treatments were different. We added this information in the main text (please see line 489-493) and we also provided the results of anova-test in supplementary Table S2.

> **"As the values of $COE_{dNO3}$ were significantly different for the 5 continents ($p < 0.01$ Table S5, i.e., Northa America, Asia, Europe, Australia, Africa and South America), the continent mean values of the calibrated parameter $COE_{dNO3}$, were used for simulations to compare … …"**

*It would be helpful if each of the panels in Figures 3-7 show D, RMSE and the correlation. Also it would be helpful if the fertilization time is shown.*

**RE:** Thanks, done as suggested. We also provided the timing of irrigation and tillage activities. To avoid redundant information, we deleted the Table.2 of the original version of the manuscript.

Baskerville, M., Reddy, N., Ofosu, E., Thevathasan, N. V., and Oelbermann, M.: Vegetation Type Does not Affect Nitrous Oxide Emissions from Riparian Zones in Agricultural Landscapes, Environ. Manage., 67, 371-383, 10.1007/s00267-020-01419-w, 2021.

Boudsocq, S., Niboyet, A., Lata, J. C., Raynaud, X., Loeuille, N., Mathieu, J., Blouin, M., Abbadie, L., and Barot, S.: Plant Preference for Ammonium versus Nitrate: A Neglected Determinant of Ecosystem Functioning?, The American Naturalist, 180, 60-69, 10.1086/665997, 2012.

Bouwman, A. F., Boumans, L. J. M., and Batjes, N. H.: Modeling global annual $N_2O$ and NO emissions from fertilized fields, Global Biogeochemical Cycles, 16, 10.1029/2001gb001812, 2002.

Butterbach-Bahl, K., Baggs, E. M., Dannenmann, M., Kiese, R., and Zechmeister-Boltenstern, S.: Nitrous oxide emissions from soils: how well do we understand the processes and their controls?, Philosophical Transactions of the Royal Society B-Biological Sciences, 368, 10.1098/rstb.2013.0122, 2013.

Chalk, P., and Smith, C.: On inorganic N uptake by vascular plants: Can 15N tracer techniques resolve the $NH_4^+$ versus $NO_3^-$ "preference" conundrum?, Eur. J. Soil Sci., 72, 1762-1779, https://doi.org/10.1111/ejss.13069, 2021.

Cui, J., Yu, C., Qiao, N., Xu, X., Tian, Y., and Ouyang, H.: Plant preference for $NH_4^+$ versus $NO_3^-$ at different growth stages in an alpine agroecosystem, Field Crops Res., 201, 192-199, https://doi.org/10.1016/j.fcr.2016.11.009, 2017.

Cui, P., Fan, F., Yin, C., Song, A., Huang, P., Tang, Y., Zhu, P., Peng, C., Li, T., Wakelin, S. A., and Liang, Y.: Long-term organic and inorganic fertilization alters temperature sensitivity of potential $N_2O$ emissions and associated microbes, Soil Biol. Biochem., 93, 131-141, https://doi.org/10.1016/j.soilbio.2015.11.005, 2016.

Daryanto, S., Wang, L., Gilhooly, W. P., III, and Jacinthe, P.-A.: Nitrogen preference across generations under changing ammonium nitrate ratios, Journal of Plant Ecology, 12, 235-244, 10.1093/jpe/rty014, 2018.

Foley, J. A., Prentice, I. C., Ramankutty, N., Levis, S., Pollard, D., Sitch, S., and Haxeltine, A.: An integrated biosphere model of land surface processes, terrestrial carbon balance, and vegetation dynamics, Global Biogeochemical Cycles, 10, 603-628, 10.1029/96gb02692, 1996.

Gelfand, I., Shcherbak, I. I., Millar, N., Kravchenko, A. N., and Robertson, G. P.: Long-term nitrous oxide fluxes in annual and perennial agricultural and unmanaged ecosystems in the upper Midwest USA, Global Change Biol., 22, 3594-3607, 10.1111/gcb.13426, 2016.

Hoben, J. P., Gehl, R. J., Millar, N., Grace, P. R., and Robertson, G. P.: Nonlinear nitrous oxide ($N_2O$) response to nitrogen fertilizer in on-farm corn crops of the US Midwest, Global Change Biol., 17, 1140-1152, 10.1111/j.1365-2486.2010.02349.x, 2011.

Hu, H.-W., Chen, D., and He, J.-Z.: Microbial regulation of terrestrial nitrous oxide formation:

understanding the biological pathways for prediction of emission rates, FEMS Microbiol. Rev., 39, 729-749, 10.1093/femsre/fuv021, 2015.

Ito, A., Nishina, K., Ishijima, K., Hashimoto, S., and Inatomi, M.: Emissions of nitrous oxide (N2O) from soil surfaces and their historical changes in East Asia: a model-based assessment, Progress in Earth and Planetary Science, 5, 10.1186/s40645-018-0215-4, 2018.

Jenkinson, D. S., Poulton, P. R., Johnston, A. E., and Powlson, D. S.: Turnover of Nitrogen-15-Labeled Fertilizer in Old Grassland, Soil Sci. Soc. Am. J., 68, 865-875, https://doi.org/10.2136/sssaj2004.8650, 2004.

Jiang, G., Zhang, W., Xu, M., Kuzyakov, Y., Zhang, X., Wang, J., Di, J., and Murphy, D. V.: Manure and Mineral Fertilizer Effects on Crop Yield and Soil Carbon Sequestration: A Meta-Analysis and Modeling Across China, Global Biogeochemical Cycles, 32, 1659-1672, 10.1029/2018gb005960, 2018.

Kucharik, C. J., Foley, J. A., Delire, C., Fisher, V. A., Coe, M. T., Lenters, J. D., Young-Molling, C., Ramankutty, N., Norman, J. M., and Gower, S. T.: Testing the performance of a Dynamic Global Ecosystem Model: Water balance, carbon balance, and vegetation structure, Global Biogeochemical Cycles, 14, 795-825, 10.1029/1999gb001138, 2000.

LaHue, G. T., van Kessel, C., Linquist, B. A., Adviento-Borbe, M. A., and Fonte, S. J.: Residual Effects of Fertilization History Increase Nitrous Oxide Emissions from Zero-N Controls: Implications for Estimating Fertilizer-Induced Emission Factors, Journal of environmental quality, 45, 1501-1508, 10.2134/jeq2015.07.0409, 2016.

Li, C. S., Aber, J., Stange, F., Butterbach-Bahl, K., and Papen, H.: A process-oriented model of N2O and NO emissions from forest soils: 1. Model development, Journal of Geophysical Research-Atmospheres, 105, 4369-4384, 10.1029/1999jd900949, 2000.

Li, L., Zheng, Z., Wang, W., Biederman, J. A., Xu, X., Ran, Q., Qian, R., Xu, C., Zhang, B., Wang, F., Zhou, S., Cui, L., Che, R., Hao, Y., Cui, X., Xu, Z., and Wang, Y.: Terrestrial N2O emissions and related functional genes under climate change: A global meta-analysis, Global Change Biology, 26, 931-943, https://doi.org/10.1111/gcb.14847, 2020.

Liu, C., Lu, M., Cui, J., Li, B., and Fang, C.: Effects of straw carbon input on carbon dynamics in agricultural soils: a meta-analysis, Global Change Biol., 20, 1366-1381, 10.1111/gcb.12517, 2014.

McSwiney, C. P., and Robertson, G. P.: Nonlinear response of N2O flux to incremental fertilizer addition in a continuous maize (Zea mays L.) cropping system, Global Change Biol., 11, 1712-1719, https://doi.org/10.1111/j.1365-2486.2005.01040.x, 2005.

Mohamed, A., Astley, H., Kun, C., Qian, Y., Dave, C., Mikk, E., Jaak, T., M, R. R., and Pete, S.: A critical review of the impacts of cover crops on nitrogen leaching, net greenhouse gas balance and crop productivity, Global Change Biol., 25, 2019.

Monfreda, C., Ramankutty, N., and Foley, J. A.: Farming the planet: 2. Geographic distribution of crop areas, yields, physiological types, and net primary production in the year 2000, Global Biogeochemical Cycles, 22, 10.1029/2007gb002947, 2008.

Parton, W. J., Mosier, A. R., Ojima, D. S., Valentine, D. W., Schimel, D. S., Weier, K., and Kulmala, A. E.: Generalized model for N2 and N2O production from nitrification and denitrification, Global Biogeochemical Cycles, 10, 401-412, https://doi.org/10.1029/96GB01455, 1996.

Peng, C., Zhu, Q., and Chen, H.: Integrating greenhouse gas emission processes into a dynamic global vegetation model of TRIPLEX-GHG, Egu General Assembly Conference, 2013,

Philibert, A., Loyce, C., and Makowski, D.: Prediction of N2O emission from local information with

Random Forest, Environ. Pollut., 177, 156-163, 10.1016/j.envpol.2013.02.019, 2013.

Rochette, P., Liang, C., Pelster, D., Bergeron, O., Lemke, R., Kroebel, R., MacDonald, D., Yan, W., and Flemming, C.: Soil nitrous oxide emissions from agricultural soils in Canada: Exploring relationships with soil, crop and climatic variables, Agric., Ecosyst. Environ., 254, 69-81, https://doi.org/10.1016/j.agee.2017.10.021, 2018.

Sebilo, M., Mayer, B., Nicolardot, B., Pinay, G., and Mariotti, A.: Long-term fate of nitrate fertilizer in agricultural soils, Proceedings of the National Academy of Sciences of the United States of America, 110, 18185-18189, 10.1073/pnas.1305372110, 2013.

Shcherbak, I., Millar, N., and Robertson, G. P.: Global metaanalysis of the nonlinear response of soil nitrous oxide ($N_2O$) emissions to fertilizer nitrogen, Proceedings of the National Academy of Sciences of the United States of America, 111, 9199-9204, 10.1073/pnas.1322434111, 2014.

Shen, S. M., Hart, P. B. S., Powlson, D. S., and Jenkinson, D. S.: The nitrogen cycle in the broadbalk wheat experiment: 15N-labelled fertilizer residues in the soil and in the soil microbial biomass, Soil Biol. Biochem., 21, 529-533, https://doi.org/10.1016/0038-0717(89)90126-0, 1989.

Song, X., Ju, X., Topp, C. F. E., and Rees, R. M.: Oxygen Regulates Nitrous Oxide Production Directly in Agricultural Soils, Environ. Sci. Technol., 53, 12539-12547, 10.1021/acs.est.9b03089, 2019.

Stehfest, E., and Bouwman, L.: N2O and NO emission from agricultural fields and soils under natural vegetation: summarizing available measurement data and modeling of global annual emissions, Nutrient Cycling in Agroecosystems, 74, 207-228, 10.1007/s10705-006-9000-7, 2006.

Thorburn, P. J., Biggs, J. S., Collins, K., and Probert, M. E.: Using the APSIM model to estimate nitrous oxide emissions from diverse Australian sugarcane production systems, Agriculture Ecosystems & Environment, 136, 343-350, 10.1016/j.agee.2009.12.014, 2010.

Tian, H., Xu, X., Liu, M., Ren, W., Zhang, C., Chen, G., and Lu, C.: Spatial and temporal patterns of CH4 and N2O fluxes in terrestrial ecosystems of North America during 1979-2008: application of a global biogeochemistry model, Biogeosciences, 7, 2673-2694, 10.5194/bg-7-2673-2010, 2010.

Tian, H., Yang, J., Lu, C., Xu, R., Canadell, J. G., Jackson, R. B., Arneth, A., Chang, J., Chen, G., Ciais, P., Gerber, S., Ito, A., Huang, Y., Joos, F., Lienert, S., Messina, P., Olin, S., Pan, S., Peng, C., Saikawa, E., Thompson, R. L., Vuichard, N., Winiwarter, W., Zaehle, S., Zhang, B., Zhang, K., and Zhu, Q.: The Global N2O Model Intercomparison Project, Bulletin of the American Meteorological Society, 99, 1231-1251, 10.1175/bams-d-17-0212.1, 2018.

Tian, H., Yang, J., Xu, R., Lu, C., Canadell, J. G., Davidson, E. A., Jackson, R. B., Arneth, A., Chang, J., Ciais, P., Gerber, S., Ito, A., Joos, F., Lienert, S., Messina, P., Olin, S., Pan, S., Peng, C., Saikawa, E., Thompson, R. L., Vuichard, N., Winiwarter, W., Zaehle, S., and Zhang, B.: Global soil nitrous oxide emissions since the preindustrial era estimated by an ensemble of terrestrial biosphere models: Magnitude, attribution, and uncertainty, Global Change Biol., 25, 640-659, 10.1111/gcb.14514, 2019.

Wang, J., Chadwick, D. R., Cheng, Y., and Yan, X.: Global analysis of agricultural soil denitrification in response to fertilizer nitrogen, Sci. Total Environ., 616, 908-917, 10.1016/j.scitotenv.2017.10.229, 2018a.

Wang, Q., Zhou, F., Shang, Z., Ciais, P., Winiwarter, W., Jackson, R. B., Tubiello, F. N., Janssens-Maenhout, G., Tian, H., Cui, X., Canadell, J. G., Piao, S., and Tao, S.: Data-driven estimates of global nitrous oxide emissions from croplands, National Science Review, 10.1093/nsr/nwz087, 2019.

Wang, R., Pan, Z., Zheng, X., Ju, X., Yao, Z., Butterbach-Bahl, K., Zhang, C., Wei, H., and Huang, B.: Using field-measured soil N2O fluxes and laboratory scale parameterization of N2O/(N2O+N2)

ratios to quantify field-scale soil N2 emissions, Soil Biol. Biochem., 148, 107904, https://doi.org/10.1016/j.soilbio.2020.107904, 2020.

Wang, W., Dalal, R. C., Reeves, S. H., Butterbach-Bahl, K., and Kiese, R.: Greenhouse gas fluxes from an Australian subtropical cropland under long-term contrasting management regimes, Global Change Biology, 17, 3089-3101, 10.1111/j.1365-2486.2011.02458.x, 2011.

Wang, W., Hou, Y., Pan, W., Vinay, N., Mo, F., Liao, Y., and Wen, X.: Continuous application of conservation tillage affects in situ N2O emissions and nitrogen cycling gene abundances following nitrogen fertilization, Soil Biol. Biochem., 157, 108239, https://doi.org/10.1016/j.soilbio.2021.108239, 2021.

Wang, Y., Guo, J., Vogt, R. D., Mulder, J., Wang, J., and Zhang, X.: Soil pH as the chief modifier for regional nitrous oxide emissions: New evidence and implications for global estimates and mitigation, Global Change Biol., 24, E617-E626, 10.1111/gcb.13966, 2018b.

Yang, L., Zhang, X., and Ju, X.: Linkage between N2O emission and functional gene abundance in an intensively managed calcareous fluvo-aquic soil, Scientific Reports, 7, 43283, 10.1038/srep43283, 2017.

Yu, Z., and Elliott, E. M.: Probing soil nitrification and nitrate consumption using Δ17O of soil nitrate, Soil Biol. Biochem., 127, 187-199, https://doi.org/10.1016/j.soilbio.2018.09.029, 2018.

Zhang, K., Peng, C., Wang, M., Zhou, X., Li, M., Wang, K., Ding, J., and Zhu, Q.: Process-based TRIPLEX-GHG model for simulating N2O emissions from global forests and grasslands: Model development and evaluation, Journal of Advances in Modeling Earth Systems, 9, 2079-2102, 10.1002/2017ms000934, 2017.

Zhou, F., Shang, Z., Zeng, Z., Piao, S., Ciais, P., Raymond, P. A., Wang, X., Wang, R., Chen, M., Yang, C., Tao, S., Zhao, Y., Meng, Q., Gao, S., and Mao, Q.: New model for capturing the variations of fertilizer-induced emission factors of N2O, Global Biogeochemical Cycles, 29, 885-897, 10.1002/2014gb005046, 2015.

Zhou, M., Zhu, B., Wang, S., Zhu, X., Vereecken, H., and Brueggemann, N.: Stimulation of N2O emission by manure application to agricultural soils may largely offset carbon benefits: a global meta-analysis, Global Change Biol., 23, 4068-4083, 10.1111/gcb.13648, 2017.

Zhu, Q., Liu, J., Peng, C., Chen, H., Fang, X., Jiang, H., Yang, G., Zhu, D., Wang, W., and Zhou, X.: Modelling methane emissions from natural wetlands by development and application of the TRIPLEX-GHG model, Geoscientific Model Development, 7, 981-999, 10.5194/gmd-7-981-2014, 2014.